# Distillation Models are Good Samplers for Diffusion Reinforcement Learning

Zunxu Liu [* 1]  Aiqiu Wu [* 1]  Zhaofan Qiu [2]  Yingwei Pan [2]  Ting Yao [2]  Tao Mei [2]

## Abstract

We present DMSampler, a framework that accelerates online diffusion reinforcement learning by replacing expensive training-time policy rollouts with a co-evolving few-step distilled sampler. Instead of repeatedly sampling the policy model for roughly 50 denoising steps, DMSampler generates reward-evaluation samples in only 4–8 steps while periodically re-distilling the sampler from the updated policy, yielding an order-of-magnitude reduction in rollout cost. The framework alternates between two stages: an RL phase that optimizes the policy using hybrid samples from the old policy and distilled sampler, and a distillation phase that realigns the few-step sampler to the improved policy. Intuitively, the distilled sampler acts as a fast proxy for the current policy during RL, and is refreshed whenever the policy improves so that sampling remains both efficient and aligned. Two designs make this loop stable and effective: hybrid distillation sampling preserves on-policy structure during rollout, and reward-aware distillation reuses high-reward trajectories to reduce forgetting during compression. Experiments on text-to-image and text-to-video generation show that DMSampler improves OCR, GenEval, and VBench performance while substantially reducing GPU hours, and that the same idea can be combined with multiple diffusion RL optimizers. Our code will be available at: https://github.com/HiDream-ai/DMSampler.

## 1. Introduction

Recent years have witnessed the remarkable success of diffusion models in high-fidelity content generation, spanning image and video synthesis (Esser et al., 2024; BlackForest, 2024; Chen et al., 2025d;a; Cai et al., 2025; Chen et al.,

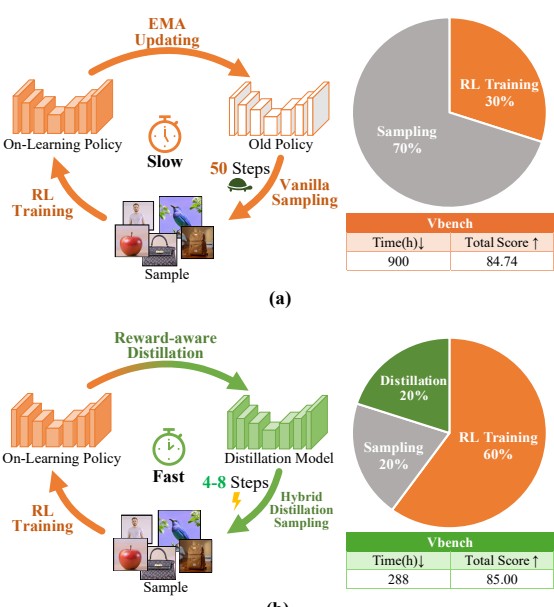

*Figure 1.* The conceptual comparison between (a) conventional Diffusion RL framework and (b) our DMSampler.

2025c; Kong et al., 2024; Peng et al., 2025; Wan et al., 2025; Gao et al., 2025; Cai et al., 2026). To align these generative models with complex, non-differentiable human preferences, such as aesthetic quality, stylistic adherence, or textual fidelity, reinforcement learning (RL) has emerged as a powerful fine-tuning paradigm. By optimizing a reward function that reflects human or automated evaluation metrics, diffusion RL methods (Wallace et al., 2024; Liu et al., 2025; Li et al., 2025; Zheng et al., 2026) can steer pretrained generative models toward higher-quality and more controllable outputs.

Despite its promise, the application of RL to large-scale diffusion models is fundamentally bottlenecked by the prohibitive sampling cost during training. Unlike conventional RL where action sampling is cheap, each policy evaluation in diffusion RL requires a full denoising trajectory, typically involving around 50 sequential steps for a single image or video. As a result, the vast majority of training time is consumed by sampling, rendering the fine-tuning process computationally expensive. Prior attempts to mitigate this issue, such as reducing the number of denoising steps during training (Liu et al., 2025; Li et al., 2025; Zheng et al., 2026), inevitably degrade sample quality and distort the training

*Equal contribution [1]University of Science and Technology of China, Hefei, China [2]HiDream.ai Inc., Beijing, China. Correspondence to: Zhaofan Qiu <zhaofanqiu@gmail.com>.

*Proceedings of the 43rd International Conference on Machine Learning*, Seoul, South Korea. PMLR 306, 2026. Copyright 2026 by the author(s).

distribution, leading to suboptimal policy convergence.

A promising pathway forward is offered by recent advances in diffusion model distillation. These techniques can compress the generative process of a diffusion model into a few-step model (typically 4–8 steps) that maintains high visual fidelity. Employing such a fast distillation model as the sampler within a diffusion RL loop presents a compelling three-fold advantage: 1) High-speed: Replacing 50-step sampling with 4-step sampling yields an order-of-magnitude acceleration in training, directly attacking the core efficiency bottleneck. 2) No-CFG: High-quality distilled models often do not require classifier-free guidance (CFG) to perform well. Their use as a sampler eliminates the distributional shift introduced by CFG, allowing the RL policy to be optimized under an unbiased objective. 3) High-quality: Counter-intuitively, the distillation model can result in straighter, more deterministic denoising trajectories. This reduction in cumulative inference error means the fast sampler can produce outputs that match or, in some cases, surpass the quality of the original multi-step sampler (Gu et al., 2025).

However, naively integrating a static distilled sampler into an RL loop is insufficient. The rapidly improving RL policy creates distributional drift, breaking the on-policy learning assumption. Therefore, we introduce **DMSampler**, a unified framework that integrates a trainable distillation model as a fast, high-quality, and adaptive sampling engine for diffusion RL. As shown in Figure 1, the core of DMSampler is a dual iterative training scheme, where the RL policy and the distillation model are alternately optimized to convergence—allowing each to stabilize before serving the other. This scheme is augmented with two key technical innovations: a hybrid distillation sampling module that strategically blends prediction from both models to maintain distributional alignment, and a reward-aware distillation objective that prioritizes high-reward trajectories to prevent forgetting. Together, these components enable DMSampler to achieve order-of-magnitude training speedups while ensuring unbiased optimization, ultimately superior performances over the prior RL-trained diffusion models. In addition, the co-evolving distillation model also show superior performance over the basic distillation methods.

The main contribution of this work is summarized as follows. We identify the bottlenecks of sampling efficiency in current diffusion RL methods, highlighting the unrealized potential of distillation models as a solution. The proposed DMSampler is the first framework to leverage a co-evolving distillation model as a trainable sampler for diffusion RL. This solution also leads to an elegant view of how to produce high-quality samples for policy optimization from distillation models, and how to synchronously update the distillation models during RL process.

## 2. Related Works

**Diffusion Model.** Diffusion models have demonstrated strong performance in image (Ho et al., 2020; Rombach et al., 2022; Podell et al., 2023; Chen et al., 2024b; Yao et al., 2025) and video generation (Singer et al., 2022; Ho et al., 2022b; He et al., 2022; Ho et al., 2022a; Guo et al., 2023; Chen et al., 2024a; Zhang et al., 2024; Long et al., 2024; Yang et al., 2024b; Peng et al., 2025) by learning data distributions through iterative denoising. Unlike early diffusion models trained with noise prediction objectives, recent models (Wan et al., 2025; Esser et al., 2024; Kong et al., 2024; Zhang et al., 2025; Gao et al., 2025; Cai et al., 2026) adopt flow matching (Liu et al., 2022), which directly learns a continuous mapping from the noise distribution to the data distribution. Despite their success, diffusion models inherently rely on sampling procedures involving multiple denoising steps, resulting in computational overhead. To address this limitation, a growing direction has focused on diffusion distillation techniques (Yin et al., 2024a;b; Salimans & Ho, 2022; Lin et al., 2025a; Sauer et al., 2024b; Luo et al., 2025; Lu et al., 2025a; Ding et al., 2025; Mao et al., 2025; Sauer et al., 2024a), which aim to compress multi-step diffusion processes into few-step generators while preserving generation quality.

**Reinforcement Learning for Diffusion Models.** To enhance diffusion models' generative capabilities and align with human preferences, research has applied RL to their post-training. Early efforts include Proximal Policy Optimization (PPO) (Black et al., 2023; Fan et al., 2023), Reward Weighted Regression (RWR) (Furuta et al., 2024; Lee et al., 2023) and Direct Reward Backpropagation fine-tuning (Clark et al., 2023). Subsequently, Diffusion-DPO (Wallace et al., 2024; Yang et al., 2024a) integrated Direct Preference Optimization (DPO) into Diffusion RL, using off-policy data to maximize positive/negative sample preference gaps. More recently, inspired by LLM progress (Guo et al., 2025; Yu et al., 2025; Yan et al., 2025), studies have explored using efficient online RL algorithms for diffusion models. Group-Relative Policy Optimization (GRPO) and its variants (Liu et al., 2025; Xue et al., 2025; Li et al., 2025) achieve strong performance by reformulating the ODE sampler into an SDE process and leveraging group reward computation in diffusion models. DiffusionNFT (Zheng et al., 2026) introduces a negative-aware fine-tuning objective, achieving strong performance and fast convergence.

Our method also operates within the online diffusion RL paradigm. Unlike prior works that are fundamentally constrained by the prohibitive sampling cost of diffusion models, we propose a novel solution: a trainable, co-evolved distillation model that serves as the adaptive sampling engine. Through a series of technical innovations, our framework fully capitalizes on the potential of this approach, establish-

ing a new and effective paradigm for efficient diffusion RL and opening up promising avenues for future research.

## 3. DMSampler

### 3.1. Overall Architecture

The overall architecture of DMSampler is illustrated in Figure 2. A fundamental departure from conventional RL lies in our use of a separate distillation model as the sampler. The critical challenge in this design is the potential divergence between the policy model and the sampling model. During policy optimization, if the sampling model remains static, a growing distributional gap emerges as the policy rapidly improves. Consequently, samples from the distillation model become less representative of the current policy, causing the on-policy learning process to partially degenerate into off-policy learning and destabilizing convergence.

To mitigate this issue, we propose a dual iterative training scheme that facilitates the co-evolution of the policy model ($\mathbf{v}_\theta$) and the distillation sampler ($\mathbf{v}_{\mathrm{dis}}$). This cyclic process alternates between two phases:

(1) **Reinforcement Learning Phase**: The distillation sampler is fixed to provide efficient sample generation. The collected samples are then used to optimize the policy model.

(2) **Distillation Phase**: The updated policy model serves as the teacher to re-distill the sampling model. This phase focuses on realigning the distillation model's output distribution with that of the current policy.

By iterating between two phases, DMSampler enables the synchronous and accelerated optimization of both models.

### 3.2. Reinforcement Learning Phase

In this phase, we adopt DiffusionNFT (Zheng et al., 2026) as the underlying RL optimizer. Its primary advantage is the elimination of explicit probability density modeling, a requirement in methods such as Flow-GRPO (Liu et al., 2025). Instead, DiffusionNFT directly utilizes online-generated positive and negative samples to optimize the flow-matching vector field via a contrastive objective. This design enables more efficient reward utilization, leading to faster convergence and more stable policy improvement.

Following the standard DiffusionNFT setup, we maintain two network instances: a trainable policy network $\mathbf{v}_\theta$ and an old network $\mathbf{v}_{\mathrm{old}}$, which is an exponential moving average (EMA) of $\mathbf{v}_\theta$ and provides stable sample generation. While the original DiffusionNFT uses only $\mathbf{v}_{\mathrm{old}}$ for sampling, DMSampler introduces a hybrid distillation sampling strategy that combines $\mathbf{v}_{\mathrm{old}}$ with a distillation model $\mathbf{v}_{\mathrm{dis}}$. We formalize this as a sampling function $f(\mathbf{v}_{\mathrm{old}}, \mathbf{v}_{\mathrm{dis}})$, whose specific design is detailed below.

Given a text prompt $c$, we generate a set of samples $\{\mathbf{x}_0\}$ via $f$ and evaluate their rewards $\{r(\mathbf{x}_0, c)\}$ using a reward model. The optimality probability $\mathbf{r}$—the probability that a sample belongs to the high-reward set—is computed using a clipped and normalized reward:

$$\mathbf{r} = \frac{1}{2}\left(\mathrm{clip}\left(\frac{r(\mathbf{x}_0, c) - \mu_{\mathcal{X}_c}}{\sigma_{\mathcal{X}_c}}, -1, 1\right) + 1\right), \quad (1)$$

where $\mu_{\mathcal{X}_c}$ and $\sigma_{\mathcal{X}_c}$ are the mean and standard deviation of rewards within the prompt group $\mathcal{X}_c$, and clip confines values to the range $[-1, 1]$.

For each sample, we obtain a noisy version $\mathbf{x}_t = (1 - \sigma_t)\mathbf{x}_0 + \sigma_t\boldsymbol{\varepsilon}$, where $\boldsymbol{\varepsilon} \sim \mathcal{N}(0, I)$ and $\sigma_t$ is a scheduled noise level. We then define an implicit positive policy $\mathbf{v}_\theta^+$ and an implicit negative policy $\mathbf{v}_\theta^-$, which represent the optimization directions for high- and low-reward samples, respectively:

$$\begin{aligned}
\mathbf{v}_\theta^+(\mathbf{x}_t, c) &= (1 - \beta)\mathbf{v}_{\mathrm{old}}(\mathbf{x}_t, c) + \beta\mathbf{v}_\theta(\mathbf{x}_t, c), \\
\mathbf{v}_\theta^-(\mathbf{x}_t, c) &= (1 + \beta)\mathbf{v}_{\mathrm{old}}(\mathbf{x}_t, c) - \beta\mathbf{v}_\theta(\mathbf{x}_t, c),
\end{aligned} \quad (2)$$

where $\beta$ is a hyperparameter controlling the divergence between the two directions.

The final DiffusionNFT loss is a reward-weighted combination of the positive and negative policy objectives:

$$\begin{aligned}
\mathcal{L}_{\mathrm{NFT}} = \mathbb{E}_{c, \mathbf{x}_0, t}\Big[&\mathbf{r}\left\|\mathbf{v}_\theta^+(\mathbf{x}_t, c) - \mathbf{v}\right\|_2^2 \\
&+ (1 - \mathbf{r})\left\|\mathbf{v}_\theta^-(\mathbf{x}_t, c) - \mathbf{v}\right\|_2^2\Big].
\end{aligned} \quad (3)$$

where $\mathbf{v}$ is the ground-truth velocity.

The core contribution of this phase is the **hybrid distillation sampling** function $f$, which resolves the online sampling bottleneck. Its goal is to leverage the speed of the few-step distillation model while ensuring the samples remain representative of the current policy $\mathbf{v}_\theta$. We systematically explore four sampling strategies to identify the optimal trade-off between efficiency and alignment:

(1) Vanilla Sampling (**S1**): Uses the original policy model $\theta_{\mathrm{old}}$ with a full denoising schedule ($T$ steps) and CFG:

$$f_{S1} = \prod_{t=1}^{T}\mathbf{v}_{\mathrm{old}}(x_t), \quad (4)$$

where $\prod$ represents the multi-step sampling process using the output of diffusion models.

(2) Full Distillation Sampling (**S2**): Relies entirely on the distillation model $\theta_{\mathrm{dis}}$ for few-step $T_d$, CFG-free sampling:

$$f_{S2} = \prod_{t=1}^{T_d}\mathbf{v}_{\mathrm{dis}}(x_t). \quad (5)$$

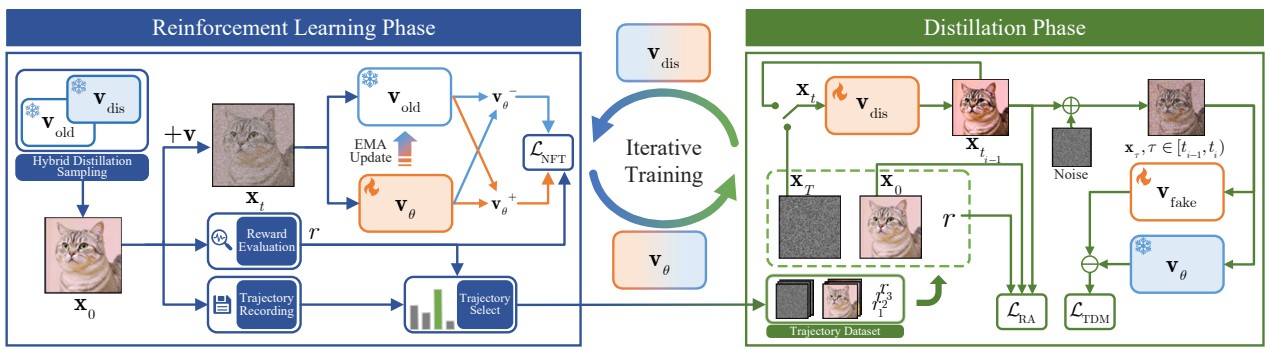

*Figure 2.* Overview of DMSampler. The framework keeps a trainable policy model $\mathbf{v}_\theta$, its EMA copy $\mathbf{v}_{\text{old}}$, and a few-step distilled model $\mathbf{v}_{\text{dis}}$ that co-evolve through alternating stages. **Left: RL phase.** With $\mathbf{v}_{\text{dis}}$ fixed, DMSampler performs hybrid distillation sampling: the old policy first handles early denoising steps to preserve the current policy distribution, and the distilled sampler then completes the remaining few steps for efficient reward evaluation. The resulting samples are scored by the reward model and used to update $\mathbf{v}_\theta$. **Right: distillation phase.** The updated policy model serves as the teacher to re-distill the model $\mathbf{v}_{\text{dis}}$ using a reward-aware objective, ensuring alignment with the current high-reward policy.

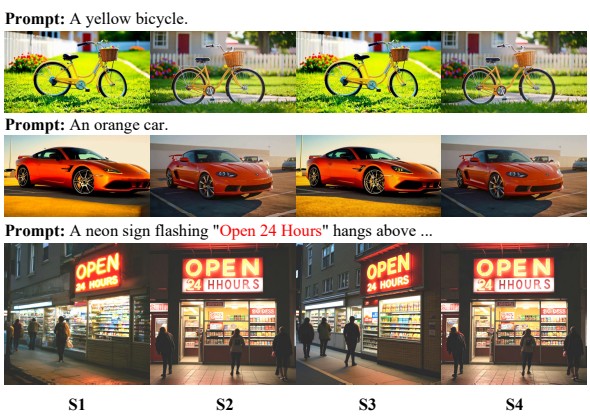

*Figure 3.* A comparison of different sampling methods.

(3) Late Distillation Sampling (**S3**): A hybrid approach where the policy model $\theta_{\text{old}}$ handles early denoising steps to establish high-level structure and semantics, after which the distillation model $\theta_{\text{dis}}$ refines the details:

$$f_{S3} = \left(\prod_{t=T_{ps}+1}^{T_{ps}+T_{ds}} \mathbf{v}_{\text{dis}}\right) \circ \left(\prod_{t=1}^{T_{ps}} \mathbf{v}_{\text{old}}\right)(x_T), \qquad (6)$$

where $T_{ds}$, and $T_{ps}$ denote the sampling steps for distillation model and policy model, respectively.

(4) Early Distillation Sampling (**S4**): Inverts the order: the distillation model performs early denoising, followed by refinement from the policy model:

$$f_{S4} = \left(\prod_{t=T_{ds}+1}^{T_{ds}+T_{ps}} \mathbf{v}_{\text{old}}\right) \circ \left(\prod_{t=1}^{T_{ds}} \mathbf{v}_{\text{dis}}\right)(x_T). \qquad (7)$$

We compare these four strategies under fixed prompts and initial noise (Figure 3), yielding three key observations: (i) S2 outputs diverge significantly from the policy model's distribution. Since the static distillation model cannot track the rapidly updating policy, using S2 alone introduces severe distributional shift. (ii) S3 maintains high fidelity to the

policy distribution. By letting the policy model define the initial trajectory, S3 preserves the on-policy characteristic necessary for stable RL, mitigates multi-step error accumulation, and still provides substantial speedup. (iii) S4 remains dominated by the distillation model's biases, as critical structural decisions are made early by $\mathbf{v}_{\text{dis}}$. Subsequent refinement by $\mathbf{v}_{\text{old}}$ cannot fully correct this initial mismatch.

Based on these findings, we adopt Late Distillation Sampling (S3) as the sampling strategy in DMSampler. It optimally balances the need for high-speed sampling with the requirement of staying aligned to the policy distribution.

### 3.3. Distillation Phase

For distillation phase, we propose **reward-aware distillation** to distill the RL-optimized policy model $\mathbf{v}_\theta$ into the few-step sampler $\mathbf{v}_{\text{dis}}$. Traditional distillation techniques typically focus on minimizing the reconstruction error of the final output, while leaving the intermediate steps of trajectory unexplored. This may induce a discrepancy between the multi-step teacher and distillation models, thereby degrading the performance of hybrid distillation model sampling. Therefore, we adopt Trajectory Distribution Matching (TDM) (Luo et al., 2025) as our distillation paradigm, which enforces alignment across the entire generation trajectory between the teacher and distillation models. Specifically, we minimize the sequence of marginal reverse KL divergences between the distillation model distribution $p_{\mathbf{v}_{\text{dis}}, t_i}(x_{t_i})$ and the corresponding policy model distribution $p_{\mathbf{v}_\theta, t_i}(x_{t_i})$ at each timestep $t_i$, which can be formulated as:

$$\mathcal{L}_{\text{TDM}} = \sum_{i=0}^{K-1} \text{KL}\big(p_{\mathbf{v}_{\text{dis}}, t_i}(x_{t_i}) \,\|\, p_{\mathbf{v}_\theta, t_i}(x_{t_i})\big), \qquad (8)$$

Since directly minimizing these distributional divergences is intractable, we employ an adversarial formulation by introducing a trainable fake score estimator alongside the policy model (i.e., the real score estimator). These two score esti-

mators respectively provide the gradients of the distillation model distribution and the policy model distribution, enabling computation of TDM loss gradients for updating the distillation model. Specific implementation details of TDM are provided in the appendix.

While TDM effectively compresses the multi-step sampling trajectories into a few-step distillation model, the distillation process is prone to mode collapse (Yin et al., 2024b; Lu et al., 2025a), which may dilute the high-reward capabilities of RL-optimized policy model. To address this, a novel reward-aware strengthening strategy is introduced to explicitly anchor the distillation to trajectories associated with superior rewards. This strategy reuses the trajectories already collected during the RL phase. Specifically, for each prompt $c_j$, we record the clean samples along with their corresponding initial noisy state and normalized reward $\mathbf{r}$, denoted as $\mathcal{C}_j = \{(\mathbf{x}_0^{(j,m)}, \mathbf{x}_T^{(j,m)}, \mathbf{r}^{(j,m)})\}_{m=1}^M$. A pre-defined threshold $\tau_{\mathrm{RA}}$ is then applied to select the high-reward trajectories within each set. If no trajectory exceeds the threshold, the one with the maximum reward is retained to avoid degradation of generalizability. The collected high-reward trajectory dataset can be denoted as:

$$\mathcal{D}_{\mathrm{RA}} = \bigcup_j \left\{ (\mathbf{x}_0^{(j,m')}, \mathbf{x}_T^{(j,m')}, \mathbf{r}^{(j,m')}) \in \mathcal{C}_j \; \right| \tag{9}$$
$$\mathbf{r}^{(j,m')} \geq \tau_{\mathrm{RA}} \text{ or } m' = \arg\max_m \mathbf{r}^{(j,m)} \right\}$$

Building upon this dataset, the distillation model is encouraged to align with the high-reward trajectories on reward-critical regions via reward-weighted trajectory loss:

$$\mathcal{L}_{\mathrm{RA}} = \mathbb{E}_{t, \{\mathbf{x}_0, \mathbf{x}_T, \mathbf{r}\} \sim \mathcal{D}_{\mathrm{RA}}}[w(\mathbf{r})||\mathbf{v}_{\mathrm{RA}} - \mathbf{v}_{\mathrm{dis}}(\mathbf{x}_t, t)||_2^2], \tag{10}$$

where $\mathbf{v}_{\mathrm{RA}} = \mathbf{x}_T - \mathbf{x}_0$ represents the target velocity, $\mathbf{x}_t = (1 - \sigma_t)\mathbf{x}_0 + \sigma_t \epsilon$ is an intermediate noisy state synthesized along a high-reward trajectory. The weighting function $w(\mathbf{r}) = 1 + \alpha \mathbf{r}$ is monotonically increasing with $\mathbf{r}$ to amplify the influence of trajectories that exhibit stronger task-aligned performance. Here, $\alpha$ is a scaling factor controlling the strength of reweighting.

### 3.4. Iterative Joint Evolution Strategy

In this section, we summarize the training objectives of each phase, as well as the iterative switching strategy between them. The complete optimization objective of the RL phase in DMSampler is:

$$\mathcal{L}_\theta = \mathcal{L}_{\mathrm{NFT}} + \lambda_{\mathrm{KL}}\mathcal{L}_{\mathrm{KL}}, \tag{11}$$

where $\mathcal{L}_{KL}$ denotes the KL-divergence regularization term, and $\lambda_{\mathrm{KL}}$ controls the trade-off between reward maximization and distributional regularization. In the subsequent distillation phase, the distillation model is optimized using:

$$\mathcal{L}_{\mathrm{dis}} = \mathcal{L}_{\mathrm{TDM}} + \lambda_{\mathrm{RA}}\mathcal{L}_{\mathrm{RA}}, \tag{12}$$

where $\lambda_{\mathrm{RA}}$ is a trade-off parameter that balances trajectory distribution matching and reward-guided strengthening.

To orchestrate the co-evolution of the policy model and the distillation sampler, we explore **convergence-based switching strategy**. This adaptive strategy alternates phases only when respective phases reach their respective convergence criteria. For the RL phase, convergence is indicated by a stabilized policy loss, together with unchanged reward values and KL divergence relative to the original model. For the distillation phase, convergence is determined by the stabilization of the evaluated reward.

## 4. Experiments

### 4.1. Experimental Settings

**Base Models.** To ensure a fair and direct comparison with established diffusion RL methods, we adopt two widely-used base models: Stable Diffusion 3.5 Medium (Esser et al., 2024) for image generation and Wan 2.1-1.3B (Wan et al., 2025) for video generation.

**Datasets and Tasks.** Our evaluation covers two core tasks for image and video diffusion models. (1) **Visual Text Rendering**: The models are required to generate images/videos containing specific text. Following prior work, we assess text fidelity using the minimum edit distance metric. The training and test splits are identical to those used in related diffusion RL studies (Liu et al., 2025; Zheng et al., 2026). (2) **Comprehensive Generation**: For image models, we use the GenEval benchmark (Ghosh et al., 2023), which evaluates text-to-image models across six compositional tasks. For video models, we employ VBench (Huang et al., 2024), a comprehensive suite that measures performance across 16 dimensions spanning visual quality, motion, and semantics.

**Implementation details.** DMSampler is implemented using PyTorch and the Diffusers library (Platen et al., 2022). For training efficiency, we integrate Low-Rank Adaptation (LoRA) modules with a rank of 32 into the pre-trained diffusion backbones, freezing all other parameters. The loss weighting coefficients $\lambda_{\mathrm{KL}}$ and $\lambda_{\mathrm{RA}}$ are set to $1 \times 10^{-4}$ and 0.05, respectively. We use the AdamW optimizer with a learning rate of $3 \times 10^{-4}$ for the reinforcement learning stage and $1 \times 10^{-4}$ for the distillation stage. For GenEval and VBench, we utilize the standard rule-based reward and evaluation models provided by each benchmark. In visual text rendering tasks, we additionally employ PaddleOCR (Cui et al., 2025) as both the reward and evaluation model.

### 4.2. Ablation Studies

All ablation studies are conducted on the visual text rendering tasks (image OCR and video OCR).

**The impact of sampling method.** We first evaluate dif-

*Table 1.* Performance comparisons of different sampling method in single reinforcement learning phase. The training time is measured on single NVIDIA A800 GPU.

| Method | Image OCR | | | Video OCR | | |
|---|---|---|---|---|---|---|
| | Steps | Time(h) | OCR Acc | Steps | Time(h) | OCR Acc |
| Base | - | - | 0.55 | - | - | 0.23 |
| RL (S1) | 40 | 160 | 0.96 | 50 | 512 | 0.46 |
| RL (S2) | 4 | 8 | 0.79 | 8 | 56 | 0.26 |
| RL (S3) | 7 | 16 | 0.95 | 16 | 160 | 0.42 |
| RL (S4) | 7 | 16 | 0.81 | 16 | 160 | 0.27 |

*Table 2.* Performance comparisons of policy model and distillation model after different training phases of DMSampler. The training time is measured on single NVIDIA A800 GPU.

| Phase | Image OCR | | Video OCR | |
|---|---|---|---|---|
| | Time(h) | OCR Acc | Time(h) | OCR Acc |
| Policy Base | - | 0.55 | - | 0.23 |
| Distillation Base | - | 0.42 | - | 0.21 |
| RL phase 1 | 16 | 0.95 | 160 | 0.42 |
| Dis phase 1 | 4 | 0.81 | 8 | 0.37 |
| RL phase 2 | 3 | 0.97 | 48 | 0.47 |
| Dis phase 2 | 1 | 0.84 | 2 | 0.40 |
| RL phase 3 | 1 | 0.97 | 16 | 0.50 |
| Dis phase 3 | 1 | 0.86 | 2 | 0.41 |

ferent sampling strategies for RL training. In this ablation study, the distillation model is kept frozen, and only a single phase of reinforcement learning is performed. For image generation (Table 1), sampling from the original model (**S1**) yields strong final performance (0.96 OCR Acc) but incurs a prohibitive computational cost of 160 GPU hours due to its lengthy denoising chain. Solely using a static distilled model for sampling (**S2**) dramatically accelerates the process but introduces severe distribution shift. As the base policy is updated, the static distilled sampler fails to track the evolving on-policy distribution, effectively reducing the RL process to off-policy learning with stale samples and thus limiting final performance. In contrast, **S3**, which employs the distilled model only in the later denoising steps—primarily affecting fine details—achieves a favorable balance. It maintains high sample quality while accelerating training, reaching a competitive OCR accuracy of 0.95 with a substantially reduced cost of 16 GPU hours. On the other hand, **S4**, while also combining both models, introduces the distillation model at early sampling stages. This leads to an irreversible degradation in sample quality, which explains its lower final accuracy of 0.81.

The trend is more pronounced in video generation. Sampling from the original video model (S1) requires $\sim$ 20 minutes per iteration, vastly exceeding the time for loss computation and update ($\sim$ 6 minutes), which severely limits the number of policy updates within a fixed budget. The static distilled sampler (S2) again devolves into inefficient off-policy learning. Our hybrid approach (S3) maintains a low sampling overhead while delivering robust performance (OCR Acc 0.42 after 160 GPU hours), effectively balancing efficiency and convergence.

**The impact of dual iterative training.** To enable the co-evolution of the distillation model with the updating policy, we next evaluate the dual iterative training strategy, which alternates between updating the policy model and the distillation model. As shown in Table 2, both the policy model and the distilled sampler exhibit consistent performance gains across successive iterative phases of DMSampler for image and video tasks. The experiment comprises six phases in total—three RL phases and three distillation phases—where the RL phases report the accuracy of the policy model and the distillation phases report that of the distilled sampler.

**Prompt:** A beautifully decorated birthday cake with smooth blue icing, the letters "**Happy 30th Jake**" elegantly spelled out on top ...

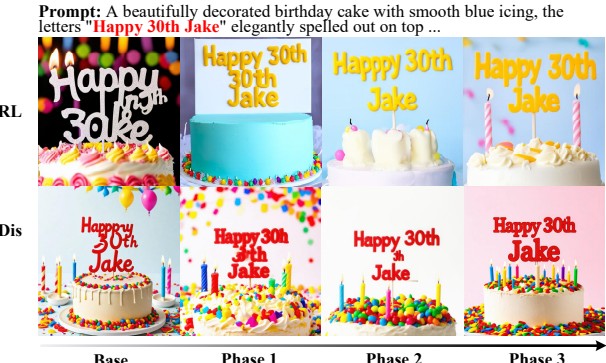

*Figure 4.* The examples of generated images after each training phase of DMSampler.

The steady improvement in both models across phases validates the effectiveness of our iterative co-training design. Furthermore, since each distillation phase requires significantly less time than an RL phase, and the majority of training time is concentrated in the first RL phase (with later phases requiring progressively fewer iterations), the overall training overhead remains manageable despite the multi-phase structure. Figure 4 showcases the images and videos generated after each training phase revealing the gradually improved quality.

**Comparison with Direct Step Reduction.** To demonstrate the advantage of using a distillation model for sampling in DMSampler, we compare it in Table 4 against an intuitive alternative: reinforcement learning methods that directly reduce the sampling (DRS) steps or disable classifier-free guidance (CFG). We evaluate three DRS variants to cover common acceleration strategies. **DRS1** reduces the sampling steps from 40 to 20 for images and from 50 to 20 for videos while keeping CFG enabled (noting that CFG doubles the sampling cost). **DRS2** builds on DRS1 by further disabling CFG. **DRS3** reduces the steps more aggressively to 10 while keeping CFG on, resulting in a sampling speed matching that of DRS2. The table compares DMSampler and these three DRS strategies on image and video OCR tasks. We also include an ablation variant, DMSampler⁻, which disables the reward-aware mechanism during the dis-

*Table 3.* Performance comparisons on GenEval benchmark. The best scores are in **bold**, and the second-best scores are underlined.

| Method | Overall | Single Object | Two Objects | Counting | Colors | Position | Attribute Binding |
|---|---|---|---|---|---|---|---|
| FLUX.1-Dev (BlackForest, 2024) | 0.66 | 0.98 | 0.81 | 0.74 | 0.79 | 0.22 | 0.45 |
| SD3.5-L (Esser et al., 2024) | 0.71 | 0.98 | 0.89 | 0.73 | 0.83 | 0.34 | 0.47 |
| Janus-Pro-7B (Chen et al., 2025b) | 0.80 | 0.99 | 0.89 | 0.59 | 0.90 | 0.79 | 0.66 |
| GPT-4o (Hurst et al., 2024) | 0.84 | 0.99 | 0.92 | 0.85 | **0.92** | 0.75 | 0.61 |
| Qwen-Image (Wu et al., 2025) | 0.91 | **1.00** | 0.95 | 0.93 | **0.92** | 0.87 | 0.83 |
| Base (SD3.5-M) (Esser et al., 2024) | 0.63 | 0.98 | 0.78 | 0.50 | 0.81 | 0.24 | 0.52 |
| FlowGRPO (Liu et al., 2025) | 0.95 | **1.00** | **1.00** | 0.95 | 0.91 | 0.94 | **0.87** |
| DiffusionNFT (Zheng et al., 2026) | 0.95 | **1.00** | **1.00** | 0.97 | **0.92** | 0.97 | 0.83 |
| **DMSampler** | **0.96** | **1.00** | **1.00** | **0.98** | 0.90 | **0.99** | 0.86 |

*Table 4.* Performance comparisons with direct step reduction. The training time is measured on single NVIDIA A800 GPU.

| Method | CFG | Image OCR | | | Video OCR | | |
|---|---|---|---|---|---|---|---|
| | | Steps | Time(h) | OCR Acc | Step | Time(h) | OCR Acc |
| Base | - | - | - | 0.55 | - | - | 0.23 |
| RL Full | ✓ | 40 | 256 | 0.96 | 50 | 512 | 0.46 |
| RL DRS1 | ✓ | 20 | 112 | 0.94 | 20 | 256 | 0.41 |
| RL DRS2 | ✗ | 20 | 72 | 0.92 | 20 | 128 | 0.31 |
| RL DRS3 | ✓ | 10 | 56 | 0.89 | 10 | 128 | 0.23 |
| DMSampler⁻ | ✗ | 7 | 25 | 0.96 | 16 | 234 | 0.48 |
| DMSampler | ✗ | 7 | 25 | 0.97 | 16 | 234 | 0.50 |

tillation phase, to validate its contribution.

Overall, DMSampler achieves higher performance than all DRS variants while being consistently faster. We observe that simple step reduction (DRS1–3) leads to a modest performance drop for image OCR (0.02–0.07 in OCR Acc) but a substantial decline for video OCR (0.05–0.23 in OCR Acc). This disparity arises because reducing steps or removing CFG degrades image quality only mildly, whereas the same operations severely harm video generation quality. Lower-quality samples inevitably hamper RL training effectiveness. In contrast, DMSampler accelerates training via a distillation model that co-evolves with the policy through iterative training, thereby maintaining high sample quality and ultimately leading to better final performance. Furthermore, comparison with DMSampler⁻ confirms the importance of reward-aware distillation, which brings clear gains (0.01 for image OCR, 0.02 for video OCR). We attribute this to the more stable positive feedback loop created by reward-aware distillation, which ensures consistent reward improvement and makes the iterative co-training scheme more effective. Figure 13 provides a visual comparison between samples generated by DRS and our hybrid distillation sampling. DRS leads to more noticeable quality degradation, which aligns with its inferior RL performance.

### 4.3. Evaluation on Comprehensive Generation

We evaluate DMSampler on the comprehensive image benchmark GenEval and the video benchmark VBench. Compared to single-reward tasks like visual text rendering, these multi-reward benchmarks pose a greater challenge to the stability of diffusion RL methods by requiring balanced optimization across diverse objectives.

**Prompt:** Drone view of waves crashing against the rugged cliffs ...

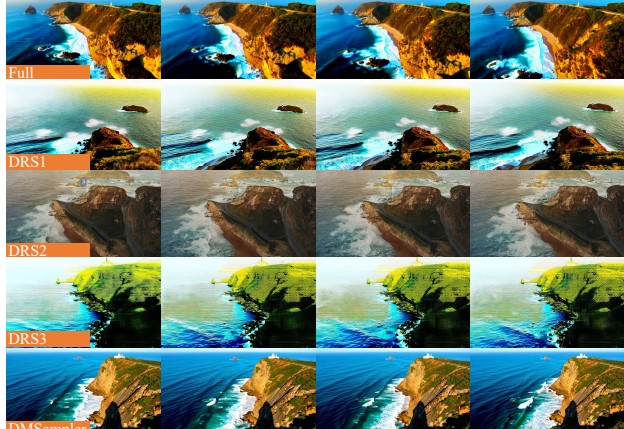

*Figure 5.* The examples of generated videos with different sampling speed-up strategy.

As shown in Table 3, DMSampler achieves the highest overall score (0.96) on GenEval, surpassing state-of-the-art diffusion RL methods FlowGRPO and DiffusionNFT (each 0.95). Compared to the base model SD3.5-M (score 0.63), DMSampler improves the overall score to 0.96, demonstrating its effectiveness in multi-reward RL scenarios. This performance also exceeds that of more advanced non-RL base models such as GPT-4o (0.84) and Qwen-Image (0.91). In terms of training efficiency on GenEval, DMSampler requires only 360 GPU hours, which is significantly lower than the baseline methods FlowGRPO (2000 hours) and DiffusionNFT (480 hours), highlighting its ability to overcome the fundamental speed bottleneck in RL training.

On the VBench benchmark, training is notably complex and computationally demanding. As presented in Table 5, DMSampler steadily improves performance throughout training. After fine-tuning, the total score of Wan 2.1-1.3B increases from 81.23 to 85.00, exceeding not only its 14B-parameter counterpart (score 83.69) but also other leading models such as CausVid and HunyuanVideo. While existing diffusion RL baselines, constrained by training duration, have not reported results on VBench, we reproduced the DiffusionNFT method as a reference. Due to the high sensitivity of video generation to sample quality discussed earlier, we configured DiffusionNFT to use a 50-step sampling procedure with CFG. Faster sampling configurations were attempted but often failed to converge on VBench due to severely de-

*Table 5.* Performance comparisons with state-of-the-art text-to-video generation models on VBench benchmark. The best scores are in **bold**, and the second-best scores are underlined. There are 16 dimensions in total, including Subject Consistency (**SC**), Background Consistency (**BC**), Temporal Flickering (**TF**), Motion Smoothness (**MS**), Dynamic Degree (**DD**), Aesthetic Quality (**AQ**), Imaging Quality (**IQ**), Object Class (**OC**), Multiple Objects (**MO**), Human Action (**HA**), Color (**C**), Spatial Relationship (**SR**), Scene (**S**), Appearance Style (**AS**), Temporal Style (**TS**), Overall Consistency (**OC'**).

| Methods | Total | SC | BC | TF | MS | DD | AQ | IQ | OC | MO | HA | C | SR | S | AS | TS | OC' |
|---|---|---|---|---|---|---|---|---|---|---|---|---|---|---|---|---|---|
| Open-Sora-2.0 (Peng et al., 2025) | 81.71 | **98.75** | 98.00 | 99.40 | **99.49** | 20.74 | 64.33 | 65.62 | 94.50 | 77.72 | 95.40 | 85.98 | 76.18 | 52.71 | 22.98 | 25.91 | 27.57 |
| CogVideoX-5B (Yang et al., 2024b) | 81.91 | 96.45 | 96.71 | 98.97 | 97.20 | 69.51 | 61.88 | 63.33 | 85.07 | 63.94 | 98.60 | 83.03 | 68.91 | 51.96 | **24.98** | 25.42 | **27.65** |
| CogVideoX1.5-5B (Yang et al., 2024b) | 82.01 | 96.56 | 96.81 | 98.53 | 98.15 | 56.16 | 62.07 | 65.34 | 83.42 | 67.28 | 97.60 | 88.40 | 79.43 | 53.28 | 24.68 | 25.42 | 27.42 |
| Hunyuan Video (Kong et al., 2024) | 83.43 | 97.22 | 97.60 | 99.39 | 99.05 | 71.94 | 60.28 | 67.24 | 83.48 | 66.71 | 94.40 | **89.79** | 72.13 | 54.46 | 22.21 | 24.52 | 26.95 |
| CausVid (Yin et al., 2025) | 83.88 | 98.09 | 97.41 | 96.89 | 97.81 | 80.60 | 64.70 | 69.67 | 92.80 | 71.68 | **99.80** | 80.34 | 64.77 | **55.84** | 24.18 | 25.29 | 27.53 |
| Wan 2.1-14B (Wan et al., 2025) | 83.69 | 97.52 | **98.09** | 99.46 | 98.30 | 64.46 | **66.07** | 69.43 | 86.28 | 69.58 | 95.40 | 88.59 | 75.39 | 45.75 | 22.64 | 23.19 | 25.91 |
| Base (Wan 2.1-1.3B) | 81.23 | 93.68 | 97.20 | 99.24 | 98.24 | 59.72 | 63.88 | 64.45 | 90.98 | 71.49 | 87.00 | 79.09 | 73.99 | 46.22 | 20.42 | 23.76 | 25.65 |
| DiffusionNFT (Zheng et al., 2026) | 84.74 | 93.46 | 97.43 | 99.46 | 98.13 | 84.72 | 65.44 | 69.43 | 94.15 | 82.70 | 93.00 | 87.77 | 87.27 | 47.60 | 21.51 | 24.68 | 25.94 |
| **DMSampler** | **85.00** | 94.51 | 96.96 | **99.52** | 98.15 | **86.11** | 65.16 | **70.32** | 95.02 | 83.16 | 95.00 | 85.74 | 86.54 | 50.65 | 21.20 | **26.24** | 26.24 |

**Prompt:** a bear on the right of a zebra, front view.

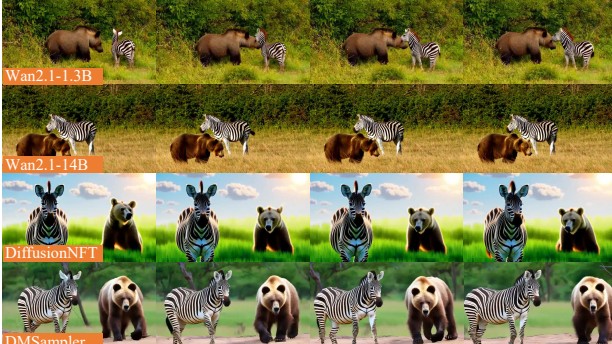

**Prompt:** A corgi is playing drum kit.

*Figure 6.* The examples of generated videos by different text-to-video models on VBench.

graded quality; a detailed analysis of this phenomenon is provided in the appendix. Even with this high-cost sampling configuration, DiffusionNFT achieves a total score of 84.74, which remains lower than DMSampler's 85.00. We attribute this performance gap primarily to the synergistic co-evolution of the policy model and the distillation model within our dual iterative training scheme, which yields benefits beyond exhaustive sampling. Overall, DMSampler not only achieves a total score improvement of $0.31\%$ over DiffusionNFT but also provides an more than 3-fold speedup, demonstrating its even greater advantage in computationally intensive video generation tasks.

Figure 6 provides a qualitative comparison of video results generated by different models on VBench, including the

*Table 6.* Performance comparisons with different diffusion RL methods on GenEval benchmark.

| Method | GPU Hours | GenEval |
|---|---|---|
| FlowGRPO | >1000 | 0.95 |
| FlowGRPO w/ DMSampler | 400 | 0.96 |
| DGPO | 240 | 0.97 |
| DGPO w/ DMSampler | 192 | 0.97 |

base Wan 2.1-1.3B, the baseline DiffusionNFT, and our DMSampler, visually demonstrating the performance gains from RL training with DMSampler. Additional generated examples and comparisons are provided in the appendix.

### 4.4. Generalization to Other RL Methods

DMSampler is not tied to DiffusionNFT. To verify its compatibility with other online diffusion RL algorithms, we integrate the same co-evolving distilled sampler with Flow-GRPO and DGPO (Luo et al., 2026) on GenEval. As shown in Table 6, adding DMSampler to FlowGRPO improves GenEval from 0.95 to 0.96 while reducing training cost from more than 1000 GPU hours to 400 GPU hours. For DGPO, DMSampler maintains the same strong GenEval score of 0.97 while reducing the training cost from 240 GPU hours to 192 GPU hours. These results indicate that DMSampler acts as a general sampling acceleration module rather than an optimizer-specific modification.

### 4.5. Out-of-Domain Evaluation

We further evaluate whether the reward-optimized policy overfits to the in-domain OCR reward. Besides OCR accuracy, we measure two out-of-domain quality metrics, aesthetic score and CLIP score, which are not directly optimized in the OCR experiment. Table 7 shows that DMSampler improves OCR accuracy from 0.96 to 0.97 while also increasing the aesthetic score from 4.46 to 4.75 and the CLIP score from 0.290 to 0.307. This suggests that DMSampler does not merely exploit the OCR reward; instead, the higher-quality hybrid rollouts and reward-aware distillation slightly mitigate reward hacking by preserving general visual and semantic quality.

*Table 7.* Out-of-Domain evaluation on the OCR benchmark.

| Method | OCR Acc | Aesthetic | CLIP Score |
|---|---|---|---|
| DiffusionNFT | 0.96 | 4.46 | 0.290 |
| DMSampler | 0.97 | 4.75 | 0.307 |

*Table 8.* Peak VRAM on Wan 2.1-1.3B .

| Method | Peak VRAM |
|---|---|
| DiffusionNFT | 57.76G |
| DMSampler | 58.77G |

*Table 9.* Quality–efficiency trade-off of different rollout samplers under the same checkpoint.

| Sampling Method | Sampling Steps | Image OCR Acc |
|---|---|---|
| Pure multi-step | 40 | 0.97 |
| Pure TDM | 4 | 0.83 |
| Hybrid | 7 | 0.92 |

| Sampling Method | Sampling Steps | VBench |
|---|---|---|
| Pure multi-step | 50 | 85.00 |
| Pure TDM | 8 | 84.21 |
| Hybrid | 16 | 84.96 |

### 4.6. Computational Overhead

Although DMSampler introduces a distillation phase, its additional memory overhead is small in practice because both the policy and distilled sampler are trained with LoRA adapters on top of the shared frozen backbone. On Wan 2.1–1.3B, the peak VRAM increases only from 57.76G for DiffusionNFT to 58.77G for DMSampler. The main bottleneck of online diffusion RL is still gradient computation in the policy update, whereas sampling forward passes are comparatively lightweight.

### 4.7. Analysis of Hybrid Sampling

We compare the performance of pure diffusion sampling, pure TDM sampling, and our hybrid sampling under the same checkpoint. As shown in Table 9, pure TDM is fastest but can lag behind the multi-step diffusion model because the distilled sampler is only partially converged within our efficiency-oriented iterative loop. Pure diffusion sampling gives the strongest instantaneous quality, reaching 0.97 OCR accuracy and 85.00 VBench, but is too expensive for frequent online updates. Hybrid sampling combines their strengths: early denoising by the policy model preserves coarse structure and on-policy semantics, while later denoising by the distilled sampler reduces cost without substantially changing the sample distribution. Consequently, hybrid sampling is much closer to pure multi-step sampling than to pure TDM in quality, while requiring only 7 image steps or 16 video steps. This explains why S3 in Table 1 is consistently stronger than pure TDM sampling and much faster than full diffusion sampling.

### 4.8. Evaluation of the Co-Evolved Distillation Model

Although the primary goal of DMSampler is to obtain an improved policy model via RL, our iterative training framework also produces a reward-optimized few-step distillation model as a valuable by-product. In this section, we compare this co-evolved distillation model from DMSampler

*Table 10.* Performance comparisons with other distillation methods on VBench benchmark. The best scores are in **bold**, and the second-best scores are underlined.

| | Method | Total | Quality | Semantic |
|---|---|---|---|---|
| Distillation | CausVid (Yin et al., 2025) | 82.88 | 83.93 | 78.69 |
| | Self Forcing (Huang et al., 2025) | 83.80 | 84.59 | 80.64 |
| | BLADE (Gu et al., 2025) | 83.38 | 84.64 | 78.30 |
| RL+Distillation | Reward Forcing (Lu et al., 2025b) | 84.13 | 84.84 | **81.32** |
| | Ours | **84.21** | **85.60** | 78.66 |

against other representative few-step distillation methods on VBench. As indicated in Table 10, our model achieves a total score of 84.21, outperforming methods such as CausVid, Self-Forcing, and BLADE. This advantage stems from DMSampler's ability to maintain distributional alignment with the evolving policy during RL updates while incorporating reward-aware distillation to mitigate catastrophic forgetting. Moreover, DMSampler exceeds the performance of Reward-Forcing, a method that directly applies RL to optimize a distilled model. This result underscores that DMSampler is not merely a sequential combination of distillation and RL, but rather an organic co-evolution framework where both components synergistically improve.

## 5. Conclusion

In this paper, we proposed DMSampler that leverages a co-evolving distilled sampler within an RL loop, establishing a novel paradigm that decouples high-speed policy evaluation from high-fidelity generation. Our core contribution is a dual iterative training scheme that alternately optimizes the policy model and the distillation sampler, ensuring their synchronous improvement. This scheme is enhanced by two key innovations: hybrid distillation sampling and reward-aware distillation. Extensive experiments on both image and video generation benchmarks validate that DMSampler achieves state-of-the-art performance while delivering an order-of-magnitude training speedup. Notably, the framework not only produces a superior final policy but also yields a high-quality, reward-optimized few-step distillation model as a valuable by-product. We believe DMSampler provides a foundational step towards scalable and efficient alignment of large generative models, significantly lowering the computational barrier for future research and application in diffusion-based reinforcement learning.

**Acknowledgments.** This work was supported by the Key Science & Technology Project of Anhui Province No. 202523o09050002 and Beijing Nova Program No. 20240484681.

## Impact Statement

This paper presents work whose goal is to advance the field of machine learning. There are many potential societal consequences of our work, none of which we feel must be specifically highlighted here.

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

## A. Diffusion Distillation for Acceleration.

The high computational cost of diffusion models has motivated extensive research on distilling multi-step diffusion processes into few-step or even single-step generators for efficient image and video synthesis. Existing studies can be broadly categorized into three paradigms: trajectory-based methods (Luhman & Luhman, 2021; Salimans & Ho, 2022; Luo et al., 2023), adversarial training–based methods (Sauer et al., 2024b; Lin et al., 2025a), and score distillation-based methods (Yin et al., 2024a;b; Zhou et al., 2024; Luo et al., 2025).

**Trajectory-based methods** aim to accelerate diffusion models by explicitly approximating or simulating the probability-flow ODE (PF-ODE) trajectories of a multi-step teacher at the instance level. Early work (Luhman & Luhman, 2021) distill the entire iterative denoising process into a single step via direct regression, which substantially reduces inference cost but often leads to over-smoothed results due to limited capacity to capture complex diffusion trajectories. Progressive distillation (Salimans & Ho, 2022) mitigates this issue by gradually increasing the step size of the student while reducing the number of function evaluations across multiple stages. Although this strategy improves training stability, it typically relies on carefully designed multi-stage pipelines and may accumulate approximation errors during successive distillation stages. Recent works explore consistency models (Luo et al., 2023; Song et al., 2023; Lu & Song, 2024; Zheng et al., 2025b), which enforce consistency along the teacher's PF-ODE trajectory and enable few-step inference without explicitly simulating the full diffusion process. However, these methods still depend on numerical solvers over pretrained PF-ODEs and can be sensitive to numerical errors when aggressively reducing the number of sampling steps.

**Adversarial training-based methods** introduce GAN-style objectives to align the output distribution of a few-step student with that of a multi-step diffusion teacher. These methods (Sauer et al., 2024b; Lin et al., 2025a;b; Sauer et al., 2024a) typically employ a discriminator to distinguish student-generated samples from real data, optimizing the student to fool the discriminator for high-fidelity image or video generation. Despite their effectiveness, adversarial objectives often inherit the well-known instability issues of GAN training, requiring extensive stabilization techniques and careful hyperparameter tuning, which complicates optimization and limits scalability.

**Score distillation-based methods** perform distribution-level alignment by matching score functions rather than directly matching samples or relying on adversarial objectives. Distribution Matching Distillation (DMD) (Yin et al., 2024b) compresses a multi-step diffusion model into a one-step generator by minimizing an approximate KL divergence between teacher- and student-induced distributions, implemented through score function differences and stabilized with an auxiliary regression loss. While enabling extreme acceleration, DMD relies on expensive pre-computed teacher samples and regression objectives, limiting scalability and robustness on complex synthesis tasks. DMD2 (Yin et al., 2024a) alleviates these issues by removing the regression loss and adopting a two-time-scale optimization scheme, further improving training stability and sample quality. In parallel, Score Identity Distillation (SiD) (Zhou et al., 2024) proposes a data-free score distillation framework that derives training signals from score identities under semi-implicit diffusion formulations, enabling effective single-step generation using only synthesized samples. Nevertheless, these methods are primarily optimized for single-step inference, which may be insufficient for complex synthesis tasks. Few-step generation offers a more balanced trade-off between efficiency and quality by allowing additional refinement steps when needed. Trajectory Distribution Matching (TDM) (Luo et al., 2025) further extends score distillation-based methods by aligning entire sampling trajectory distributions, leading to improved robustness under few-step sampling regimes.

## B. Trajectory Distribution Matching.

We briefly introduce the procedure of TDM. The core idea of TDM is to distill a multi-step diffusion teacher $\mathbf{v}_{\text{tch}}$ into a few-step student $\mathbf{v}_{\text{stu}}$ by aligning their sampling trajectories at the distribution level. Specifically, let $\{x_{t_0}, x_{t_1}, \ldots, x_{t_K}\}, 0 = t_0 < t_1 < \cdots < t_K = T$ denote the intermediate states along the student's sampling trajectory. TDM seeks to minimize a sequence of marginal reverse KL divergences between the implicit distribution on the student trajectory $p_{\text{stu},t_i}(x_{t_i})$ that on the teacher trajectory $p_{\text{tch},t_i}(x_{t_i})$, which can be formulated as:

$$\mathcal{L}_{\text{TDM}} = \sum_{i=0}^{K-1} \text{KL}\big(p_{\text{stu},t_i}(x_{t_i}) \,\|\, p_{\text{tch},t_i}(x_{t_i})\big), \tag{13}$$

The objective requires the score function of teacher model $\nabla_{x_\tau} \log p_{\text{tch},\tau|t_i}(x_\tau)$ as well as the score of the student's implicit trajectory-induced distribution $\nabla_{x_\tau} \log p_{\text{stu},\tau|t_i}(x_\tau)$. While the former can be directly evaluated using the teacher model (i.e., the real score estimator), the latter is generally intractable. TDM resolves this by introducing an auxiliary fake score

model $\mathbf{v}_{\text{tch}}$, which provides an approximation to the student distribution score. The fake score model $\mathbf{v}_{\text{fake}}$ is trained by denoising as:

$$\mathcal{L}_{\text{fake}} = \sum_{i=0}^{K-1} \mathbb{E}_{x_{t_i} \sim p_{\text{stu},t_i}} \mathbb{E}_{x_\tau \sim q(\hat{x}_{t_i})} \left[ \|\mathbf{v}_{\text{fake}}(x_\tau, \tau) - \hat{x}_{t_i}\|_2^2 \right], \tag{14}$$

where $p_{\mathbf{v}_{stu},t_i}$ represents the student's distribution at timestep $t_i$, $\hat{x}_{t_i}$ is the clean data corresponding to $x_{t_i}$, and $x_\tau$ is a perturbed sample obtained by adding noise to $\hat{x}_{t_i}$. With score functions $s_{\text{tch}}$ and $s_{\text{fake}}$ respectively estimated by $\mathbf{v}_{\text{tch}}$ and $\mathbf{v}_{\text{fake}}$, the distilled model $\mathbf{v}_{\text{dis}}$ can be trained by backpropagating the gradient of the reverse KL divergence through the generated trajectory. For more stable knowledge transfer, the KL objective is evaluated on diffused marginal distributions, defined as: $p_{\text{stu},\tau|t_i}(\mathbf{x}_\tau) \triangleq \int q(\mathbf{x}_\tau \mid \mathbf{x}_{t_i}) p_{\text{stu},t_i}(\mathbf{x}_{t_i}) \, d\mathbf{x}_{t_i}$ with an analogous definition for $p_{\mathbf{v}_{tch},\tau|t_i}$. Under this formulation, student model parameters $\theta_{stu}$ can be approximated as:

$$\begin{aligned} \nabla_{\theta_{\text{stu}}} \mathcal{L}_{\text{TDM}} &= \sum_{i=0}^{K-1} \sum_{\tau=t_i}^{t_{i+1}} \lambda_\tau \left[ \nabla_{x_\tau} \log p_{\text{stu},\tau|t_i}(x_\tau) - s_\theta(x_\tau, \tau) \right] \frac{\partial x_{t_i}}{\partial \theta_{\text{stu}}} \\ &\approx \sum_{i=0}^{K-1} \sum_{\tau=t_i}^{t_{i+1}} \lambda_\tau \left[ s_{\text{fake}}(x_\tau) - s_\theta(x_\tau, \tau) \right] \frac{\partial x_{t_i}}{\partial \theta_{\text{stu}}}, \end{aligned} \tag{15}$$

where $\lambda_\tau$ is a weighting coefficient. The gradient is backpropagated through the student-generated trajectory $x_{t_i}$, thereby explicitly enforcing trajectory distribution alignment between the teacher and the student.

## C. Convergence-Based Switching.

The effectiveness of the proposed dual iterative training scheme critically depends on when to alternate between the RL phase and the distillation phase. To this end, we adopt a convergence-based switching strategy. The key idea lies in switching phase when the optimization in the current stage has sufficiently stabilized, thereby ensuring that (i) the policy model provides a reliable teacher for distillation, and (ii) the distillation model is sufficiently aligned before being reused for policy optimization.

**RL Phase Convergence.** During the RL phase, the distillation model $\mathbf{v}_{\text{dis}}$ is fixed, and the policy model $\mathbf{v}_\theta$ is optimized with the objective $\mathcal{L}_\theta = \mathcal{L}_{\text{NFT}} + \lambda_{\text{KL}} \mathcal{L}_{\text{KL}}$. We consider the RL phase to have converged when both the optimization loss and the achieved reward exhibit saturation behavior. Concretely, let $k$ denote the iteration index and $W$ the length of a sliding window. For a sequence $z_i$, we define its windowed average as:

$$\langle z \rangle_k \triangleq \frac{1}{W} \sum_{i=k-W+1}^{k} z_i, \quad k \geq W. \tag{16}$$

RL convergence is declared when the variation between two consecutive windows becomes sufficiently small for both the training loss and the normalized reward, i.e.,

$$|\langle \mathcal{L}_\theta \rangle_k - \langle \mathcal{L}_\theta \rangle_{k-W}| < \tau_\theta, \tag{17}$$

and simultaneously,

$$|\langle \mathbf{r} \rangle_k - \langle \mathbf{r} \rangle_{k-W}| < \tau_r, \tag{18}$$

where $\tau_\theta$ and $\tau_r$ are predefined thresholds controlling the tolerance for loss fluctuation and reward improvement, respectively. When neither the optimization objective nor the reward exhibits meaningful progress, we regard the policy model as having reached a stable solution and switch to the distillation phase.

**Distillation Convergence.** In the distillation phase, the policy model $\mathbf{v}_\theta$ is fixed. Due to the inherent instability of the distillation loss, we determine distillation convergence by directly monitoring the reward improvement of the distilled sampler. Concretely, every $W$ distillation iterations, we evaluate the distilled model on the same validation mini-batch and compute the corresponding average reward $\langle \mathbf{r}_{\text{dis}} \rangle_k$. Distillation is considered converged if the reward improvement between two consecutive evaluation windows becomes negligible:

$$|\langle \mathbf{r}_{\text{dis}} \rangle_k - \langle \mathbf{r}_{\text{dis}} \rangle_{k-W}| < \tau_{\text{dis}}, \tag{19}$$

**Prompt: A panda playing on a swing set**

*Figure 7.* Additional examples of generated images and videos under different sampling strategy.

where $\tau_{\mathrm{dis}}$ denotes a predefined reward threshold during distillation phase. Since $\mathbf{v}_{\mathrm{dis}}$ operates with only a few sampling steps, the additional cost of reward evaluation is minimal. Once convergence is detected, the distilled model is regarded as sufficiently aligned with the RL-optimized policy and is reused to initialize the subsequent RL phase.

We find the convergence-based switching criteria to be robust within a reasonable range of threshold values, and we select the final thresholds based on ablation studies.

## D. Analysis on the impact of CFG in online Diffsuion RL.

In this section, we briefly analyze the potential impact of incorporating CFG into the sampling process of online Diffusion RL from several perspectives.

1) Following (Black et al., 2023), the iterative denoising process in flow matching models can be formulated as a Markov Decision Process (MDP), denoted by the tuple $(S, A, \rho_0, P, R)$. Specifically, the state at time step $t$ is defined as $s_t \triangleq (c, t, x_t)$, the action corresponds to the denoised sample predicted by the model $a_t \triangleq x_{t-1}$, and the policy is given by $\pi(a_t \mid s_t) \triangleq p_\theta(x_{t-1} \mid x_t, c)$. However, in the presence of CFG, the policy described above effectively entails a "base prior" (also referred to as a "general action distribution") which is induced jointly by the null prompt and the model configuration, that is $\pi(a_t \mid s_t) \triangleq p_\theta(x_{t-1} \mid x_t, c, \phi)$, where $\phi$ represents the null prompt. The specificity of this issue stems from the architectural design where the conditional prediction $v_\theta(x_t, c)$ and the unconditional prediction $v_\theta(x_t, \phi)$ share the exact same set of model parameters $\theta$. This parameter coupling introduces two critical challenges during the post-training reinforcement learning phase:

a. Gradient Contamination and Conflict: The first issue concerns whether to exclude the gradients of the unconditional branch $v_\theta(x_t, \phi)$ during optimization. If the gradients from $v_\theta(x_t, \phi)$ are incorporated into the final update (as is implicit when optimizing the CFG output directly), the resulting gradient vector is contaminated by signals unrelated to the current task $c$. This leads to an undesirable gradient offset.

b. Furthermore, in multi-task post-training scenarios, this shared dependency creates a bottleneck. Gradients derived from different tasks may exhibit conflicting directions regarding the unconditional prior. For instance, one prompt might incentivize high-magnitude motion while another demands stationarity. These conflicting signals applied to the shared unconditional branch can lead to destructive interference, hampering the model's ability to learn distinct task-specific policies.

Non-stationary Baseline and Prior Drift: Even if the gradients of $v_\theta(x_t, \phi)$ are explicitly detached to prevent direct contamination, a subtler but equally pernicious issue remains. Since the parameters $\theta$ are updated to optimize the conditional branch $v_\theta(x_t, c)$, the behavior of the unconditional branch $v_\theta(x_t, \phi)$ implicitly shifts.

In the context of Classifier-Free Guidance, $v_\theta(x_t, \phi)$ serves as a reference baseline (or general action distribution). An implicit update to this baseline is analogous to a non-stationary prior. Consequently, the RL agent is forced to optimize against a moving target, which introduces significant instability and potential bias into the training process, ultimately

degrading the reliability of the guidance signal.

2) From the perspective of the loss function, we analyze the problem by considering the final loss function of Diffusion RL as an L2 loss:

$$L(\theta) = \|w(r)v_\theta(x_t, c) - w'(r)v\|_2^2 + C, \tag{20}$$

where $v$ is sampled from the old policy. The final loss functions of both DiffusionNFT and FlowGRPO can somehow be expressed in the form of the above L2 loss. With CFG sampling,

$$v = (1 + \beta)v_{\text{old}}(c) - \beta v_{\text{old}}(\phi), \tag{21}$$

then the loss turns into

$$
\begin{aligned}
\mathcal{L}(\theta) &= \|w(r)v_\theta(x_t, c) - w'(r)((1+\beta)v_{\text{old}}(x_t, c) - \beta v_{\text{old}}(x_t, \phi))\|_2^2 + C \\
&= \|(1+\beta)[w(r)v_\theta(x_t, c) - w'(r)v_{\text{old}}(x_t, c)] - \beta[w(r)v_\theta(x_t, c) - w'(r)v_{\text{old}}(x_t, \phi)]\|_2^2 + C \\
&= \underbrace{(1+\beta)^2\|w(r)v_\theta(x_t, c) - w'(r)v_{\text{old}}(x_t, c)\|_2^2}_{\text{Term I: Conditional Alignment}} \\
&\quad + \underbrace{\beta^2\|w(r)v_\theta(x_t, c) - w'(r)v_{\text{old}}(x_t, \phi)\|_2^2}_{\text{Term II: Unconditional Regularization (Harmful)}} \\
&\quad - \underbrace{2\beta(1+\beta)\langle\ldots\rangle}_{\text{Term III: Cross-Interaction}} + C.
\end{aligned}
\tag{22}
$$

**Term I** represents the desired objective: aligning the updated policy with the high-reward conditional reference. However, **Term II** introduces a pathological optimization constraint. It explicitly penalizes the distance between the current *conditional* output $v_\theta(x_t, c)$ and the *unconditional* reference $v_{\text{old}}(x_t, \phi)$.

Minimizing Term II effectively forces the conditional policy to mimic the generic, unconditioned prior. This acts as a counter-productive regularizer that suppresses task-specific features, pulling the policy towards the mean action distribution and diminishing the distinctiveness required for specific tasks. As $\beta$ increases, this "prior-seeking" behavior competes more aggressively with the task-specific objective, leading to suboptimal convergence and the potential loss of conditional control.

The second term aims to make the policy model fit the unconditional part of the old model (i.e., the irrelevant or meaningless components that should be removed), which may lead to biased optimization.

DiffusionNFT also explores the issue that using CFG sampling can complicate the training process, leading to its deactivation throughout the entire training. To mitigate the sharp decline in sample quality caused by disabling CFG, DiffusionNFT trains for 800 iterations with pickscore, clipscore, and hpsv2 as rewards to enhance image quality. Additionally, some other works (Zheng et al., 2025a) and the community have found that omitting CFG during Diffusion RL training can achieve comparable or even superior results compared to training with CFG.

## E. Analysis of sampling step.

As shown in Figure 7, for image and video models, the impact of reducing the number of sampling steps and CFG on sampling quality is not uniform. For images, even reducing the number of sampling steps to 10 can still yield a relatively reasonable sample (albeit with a decrease in quality), which is the reason why DiffusionNFT can adopt this strategy for image tasks. However, such samples still lead to poor image generation quality after training; therefore, to ensure high-quality samples, 40-step sampling is also used in DiffusionNFT's multi-reward task. For videos, directly disabling CFG or reducing the number of sampling steps (as in DiffusionNFT) will result in significant video quality degradation, and such severely degraded samples can have a negative impact on the RL process.

## F. Additional Text-to-Image Results

We further evaluate DMSampler on additional image settings to assess architectural and benchmark generalization. First, we apply DMSampler to FLUX.1-dev (11B) on the OCR benchmark. As shown in Table 11, DMSampler achieves an OCR

*Table 11.* Additional OCR results on FLUX.1-dev.

| Method | GPU Hours | OCR Acc |
|---|---|---|
| DiffusionNFT | 240 | 0.96 |
| DMSampler | 160 | 0.97 |

*Table 12.* Additional text-to-image benchmark results on HPDv2 and DrawBench.

| Method | GPU Hours | HPSv2 on HPDv2 | HPSv2 on DrawBench | PickScore on DrawBench |
|---|---|---|---|---|
| DiffusionNFT | 160 | 0.376 | 0.350 | 17.29 |
| DMSampler | 112 | 0.380 | 0.376 | 17.49 |

accuracy of 0.97 using 160 GPU hours, compared with 0.96 accuracy and 240 GPU hours for DiffusionNFT. This result shows that the proposed co-evolving sampler transfers beyond the SD3.5-Medium backbone used in the main experiments, and that its acceleration benefit remains clear on a larger image diffusion architecture.

Second, we evaluate on HPDv2 and DrawBench, using HPSv2 as the reward-side evaluation and PickScore as an out-of-domain preference metric. As summarized in Table 12, DMSampler consistently improves over DiffusionNFT while reducing GPU hours from 160 to 112. On DrawBench, DMSampler improves both the optimized HPSv2 score and the out-of-domain PickScore, indicating that the gains are not limited to GenEval's compositional rules and do not simply overfit to the in-domain reward.

## G. Limitations

DMSampler reduces the dominant sampling cost in online diffusion RL, but it also introduces additional algorithmic components, including a co-evolving distilled sampler, a distillation phase, and switching criteria between the two phases. Although our experiments show that the peak VRAM overhead is small under LoRA fine-tuning and that the switching thresholds are not highly sensitive, the training pipeline is still more complex than a single-stage RL baseline.

Another limitation is that our empirical study focuses on text-to-image and text-to-video generation with automatic reward and benchmark metrics. These evaluations cover OCR, GenEval, VBench, HPDv2, DrawBench, and out-of-domain preference metrics, but they still cannot exhaustively capture all forms of human preference alignment. Like other RL-based post-training methods, DMSampler can also be affected by reward misspecification when the reward model emphasizes only a narrow target. Designing richer multi-objective rewards and conducting broader human preference studies are important directions for future work.

## H. Convergence of DMSampler

Since both DiffusionNFT and TDM have been shown to exhibit convergence, with the convergence of their respective stages achieved within a finite number of steps, we empirically observe stable convergence behavior in DMSampler when iteratively switching between the two stages. Importantly, the objectives of the two stages are aligned: both aim to improve the reward, rather than forming an adversarial or conflicting optimization process. This objective consistency allows DMSampler to empirically converge to a high-reward state.

## I. Discussion on Reward Hacking

Reward hacking is a well-known and widely observed phenomenon in reinforcement learning, particularly when optimization is driven by a single or a small set of pre-defined reward signals. In our experiments, we observe that while optimizing task-specific rewards effectively improves the targeted metrics, certain aspects of perceptual quality may degrade. We note that this behavior is not unique to our method and has been reported in many prior works on diffusion-based and generative reinforcement learning.

Importantly, such degradation does not necessarily stem from the RL algorithm itself, but rather from the incompleteness of the reward specification. When the reward function captures only a narrow objective, aspects of quality that are not explicitly rewarded may be overlooked during optimization. In practical applications, this limitation can be mitigated by designing more comprehensive reward formulations that jointly consider multiple quality dimensions and task requirements. We view this as an orthogonal challenge related to reward design, rather than a deficiency of the proposed RL framework.

## J. Reward Curve Comparison with Direct Step Reduction

We further compare the efficiency of DMSampler against various Direct Step Reduction (DRS) strategies. As shown in Figure 8, while DRS methods gain speed by reducing sampling steps or removing CFG, they suffer from slower and limited

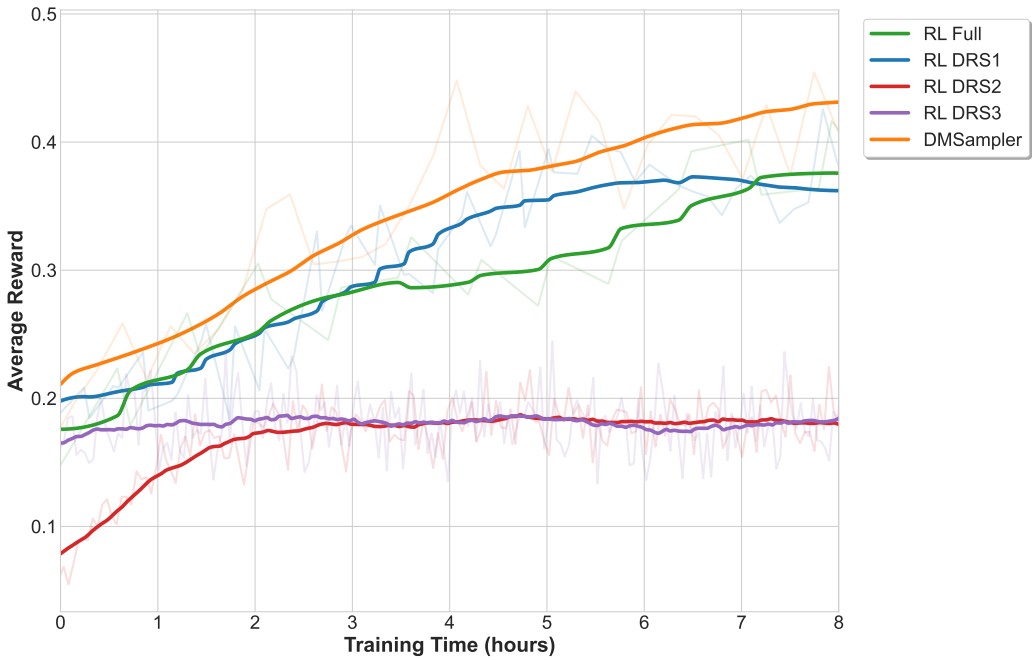

*Figure 8.* Evaluation reward curves comparison with direct step reduction on Video OCR task. The training time is measured on **16 NVIDIA A800 GPUs**.

reward growth. Specifically, all DRS variants exhibit a clear trade-off: DRS2 and DRS3 reduce per-iteration cost but quickly plateau due to low-quality trajectories, while DRS1 and full-step sampling achieve higher rewards at the cost of longer training time. In contrast, DMSampler consistently yields faster reward growth and superior performance. This is because our co-evolving distilled sampler preserves the multi-step trajectory quality while drastically cutting sampling costs, providing a stable and informative reward signal for more efficient reinforcement learning.

## K. Hyperparameters Setting

Hyperparameters used throughout this work are summarized in Table 13. All hyperparameters in DMSampler are selected based on ablation studies to identify a stable and well-performing configuration.

## L. Training Algorithm

The algorithm for DMSampler is presented in Algorithm 1.

## M. Discussion on Distillation with CFG

We observe that enabling CFG during the distillation stage provides stronger conditional teacher trajectories for few-step TDM, leading to better prompt alignment and improved distilled-sampler performance. Therefore, CFG-guided distillation is complementary to our framework, and can be beneficial when combined with RL backbones.

| Description | Parameter | Value |
|---|---|---|
| *Model Architecture* | | |
| Base diffusion model (image) | – | SD 3.5 Medium |
| Base diffusion model (video) | – | Wan 2.1–1.3B |
| LoRA rank | – | 32 |
| *Hybrid Distillation Sampling* | | |
| Vanilla sampling steps (image) | $T$ | 40 |
| Vanilla sampling steps (video) | $T$ | 50 |
| Distillation sampling steps (image) | $T_d$ | 4 |
| Distillation sampling steps (video) | $T_d$ | 8 |
| Sampling steps of policy model in Late/Early Distillation Sampling (image) | $T_{ps}$ | 4 |
| Sampling steps of policy model in Late/Early Distillation Sampling (video) | $T_{ps}$ | 12 |
| Sampling steps of distillation model in Late/Early Distillation Sampling (image) | $T_{ds}$ | 3 |
| Sampling steps of distillation model in Late/Early Distillation Sampling (video) | $T_{ds}$ | 4 |
| *Reinforcement Learning Phase* | | |
| NFT hyperparameter | $\beta$ | 0.1 |
| KL regularization weight | $\lambda_{\text{KL}}$ | $1 \times 10^{-4}$ |
| RL optimizer | – | AdamW |
| RL learning rate | – | $3 \times 10^{-4}$ |
| *Distillation Phase* | | |
| Sampling steps of distillation model (image) | $K$ | 4 |
| Sampling steps of distillation model (video) | $K$ | 8 |
| Reward threshold | $\tau_{\text{RA}}$ | 0.8 |
| Reward reweighting scale | $\alpha$ | 1.0 |
| Distillation loss weight | $\lambda_{\text{RA}}$ | 0.05 |
| Distillation optimizer | – | AdamW |
| Distillation learning rate | – | $1 \times 10^{-4}$ |
| *Iterative Switching* | | |
| Sliding window | $W$ | 20 |
| RL loss fluctuation threshold | $\tau_\theta$ | 0.001 |
| RL reward fluctuation threshold | $\tau_r$ | 0.05 |
| Distillation reward fluctuation threshold | $\tau_{\text{dis}}$ | 0.05 |

*Table 13.* Hyperparameters setting in DMSampler.

---

**Algorithm 1** Training DMSampler

---

**Require:** Pretrained base diffusion model and its corresponding distilled base model; reward model; prompt set $\mathcal{D}_c$; hybrid distillation sampling steps $(T_{ps}, T_{ds})$; thresholds $(\tau_{RA}, \tau_\theta, \tau_r, \tau_{dis})$; sliding window size $W$.

1: Initialize policy model $\mathbf{v}_\theta$ and old model $\mathbf{v}_{\text{old}}$ from the pretrained base diffusion model.
2: Initialize distillation sampler $\mathbf{v}_{\text{dis}}$ from the distilled base model; initialize fake score model $\mathbf{v}_{\text{fake}}$.
3: **while** policy model reward continues to improve **do**
4:   /* RL Phase */
5:   Set $\mathbf{v}_\theta$ trainable and freeze $\mathbf{v}_{\text{dis}}$.
6:   **repeat**
7:     Sample a mini-batch of prompts $\{c_j\} \subset \mathcal{D}_c$.
8:     Initialize per-prompt trajectory buffers $\{\mathcal{C}_j\}$ to empty.
9:     Sample initial noise $\mathbf{x}_T \sim \mathcal{N}(0, I)$.
10:     Perform hybrid distillation sampling $\mathbf{x}_0 = f_{S3}(\mathbf{v}_{\text{old}}, \mathbf{v}_{\text{dis}})$.
11:     Compute reward $r(\mathbf{x}_0, c_j)$ using the reward model, and normalize to $\mathbf{r}$.
12:     Record $(\mathbf{x}_0, \mathbf{x}_T, \mathbf{r})$ in $\mathcal{C}_j$.
13:     Sample noise level $t$ and construct $\mathbf{x}_t = (1 - \sigma_t)\mathbf{x}_0 + \sigma_t\boldsymbol{\epsilon}$, $\boldsymbol{\epsilon} \sim \mathcal{N}(0, I)$.
14:     Construct implicit positive and negative policies:
15:       $\mathbf{v}_\theta^+(\mathbf{x}_t, c_j) = (1 - \beta)\mathbf{v}_{\text{old}}(\mathbf{x}_t, c_j) + \beta\mathbf{v}_\theta(\mathbf{x}_t, c_j)$,
16:       $\mathbf{v}_\theta^-(\mathbf{x}_t, c_j) = (1 + \beta)\mathbf{v}_{\text{old}}(\mathbf{x}_t, c_j) - \beta\mathbf{v}_\theta(\mathbf{x}_t, c_j)$.
17:     Compute DiffusionNFT loss $\mathcal{L}_{\text{NFT}}$ and KL regularization loss $\mathcal{L}_{\text{KL}}$.
18:     Update $\mathbf{v}_\theta$ by minimizing $\mathcal{L}_\theta = \mathcal{L}_{\text{NFT}} + \lambda_{\text{KL}}\mathcal{L}_{\text{KL}}$.
19:     Update model $\mathbf{v}_{\text{old}}$ via EMA.
20:   **until** RL convergence.
21:   /* Distillation Phase */
22:   Freeze $\mathbf{v}_\theta$ and set $\mathbf{v}_{\text{dis}}$ trainable.
23:   Construct high-reward trajectory dataset $\mathcal{D}_{\text{RA}}$ from $\{\mathcal{C}_j\}$ via thresholding with $\tau_{RA}$.
24:   **repeat**
25:     /* Fake score model update */
26:     Sample intermediate states $\mathbf{x}_{t_i} \sim p_{\mathbf{v}_{\text{stu}}, t_i}$ along the distillation trajectory.
27:     Update $\mathbf{v}_{\text{fake}}$ by minimizing $\mathcal{L}_{\text{fake}}$.
28:     /* Trajectory Distribution Matching */
29:     Generate distillation trajectory $\{\mathbf{x}_{t_i}\}$ using $\mathbf{v}_{\text{dis}}$.
30:     Compute gradients of $\mathcal{L}_{\text{TDM}}$ using $\mathbf{v}_\theta$ and $\mathbf{v}_{\text{fake}}$.
31:     Update $\mathbf{v}_{\text{dis}}$ by minimizing $\mathcal{L}_{\text{TDM}}$.
32:     /* Reward-aware strengthening */
33:     Sample $(\mathbf{x}_0, \mathbf{x}_T, r) \sim \mathcal{D}_{\text{RA}}$ and noise level $t$, then form $\mathbf{x}_t = (1 - \sigma_t)\mathbf{x}_0 + \sigma_t\boldsymbol{\epsilon}$, $\boldsymbol{\epsilon} \sim \mathcal{N}(0, I)$.
34:     Compute reward-aware loss $\mathcal{L}_{\text{RA}}$.
35:     Update $\mathbf{v}_{\text{dis}}$ by minimizing $\mathcal{L}_{\text{dis}} = \mathcal{L}_{\text{TDM}} + \lambda_{RA}\mathcal{L}_{\text{RA}}$.
36:   **until** distillation convergence.
37: **end while**
38: **Output:** RL-optimized policy model $\mathbf{v}_\theta$ and distilled sampler $\mathbf{v}_{\text{dis}}$.

---

**Prompt: A vintage radio with a worn wooden casing, the dial glowing softly and labeled "Tune to Adventure", set against a backdrop of an old, cozy living room with a fireplace and a bookshelf.**

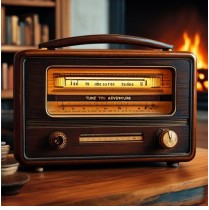 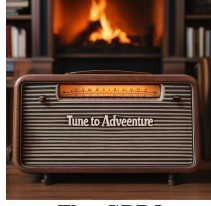 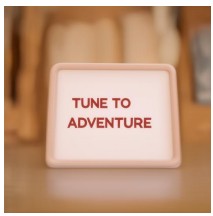 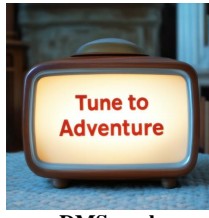

SD3.5-M      FlowGRPO      DiffusionNFT      DMSampler

**Prompt: An ancient, weathered stone tablet, partially covered in moss, with the intricate inscription "Kingdom of the Sun" prominently carved into its surface, set against a backdrop of dense, overgrown forest.**

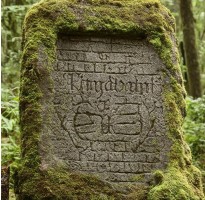 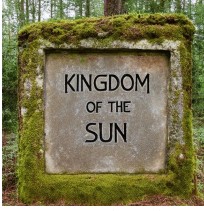 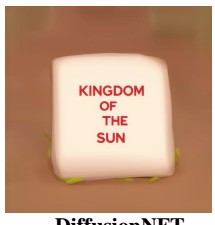 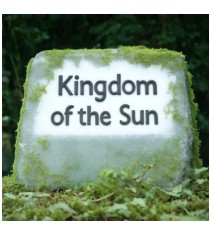

SD3.5-M      FlowGRPO      DiffusionNFT      DMSampler

**Prompt: a photo of  yellow train**

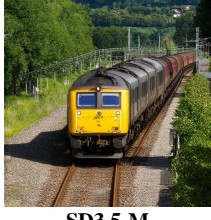 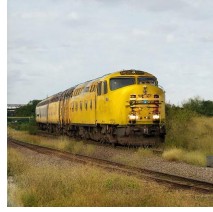 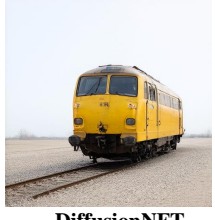 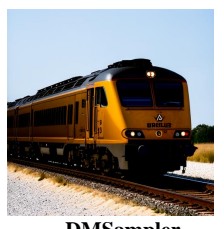

SD3.5-M      FlowGRPO      DiffusionNFT      DMSampler

**Prompt: a photo of four donuts**

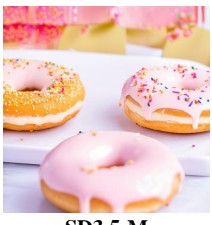 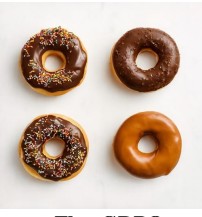 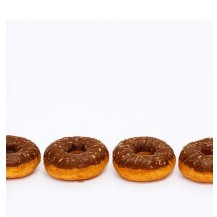 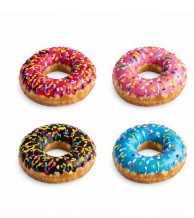

SD3.5-M      FlowGRPO      DiffusionNFT      DMSampler

**Prompt: a photo of four stop signs**

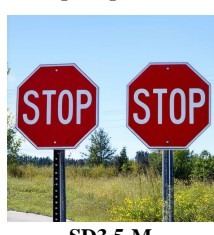 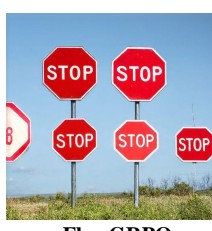 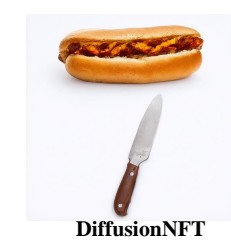 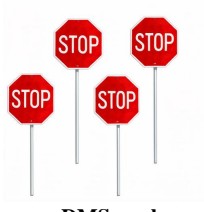

SD3.5-M      FlowGRPO      DiffusionNFT      DMSampler

**Prompt: a photo of a hot dog above a knife**

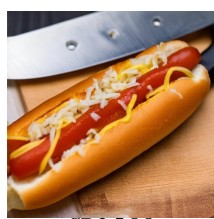 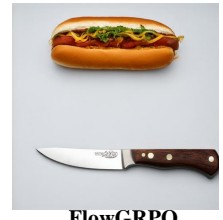 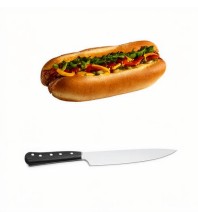

21

SD3.5-M      FlowGRPO      DiffusionNFT      DMSampler

*Figure 9.* Examples of generated images of different text-to-image models.

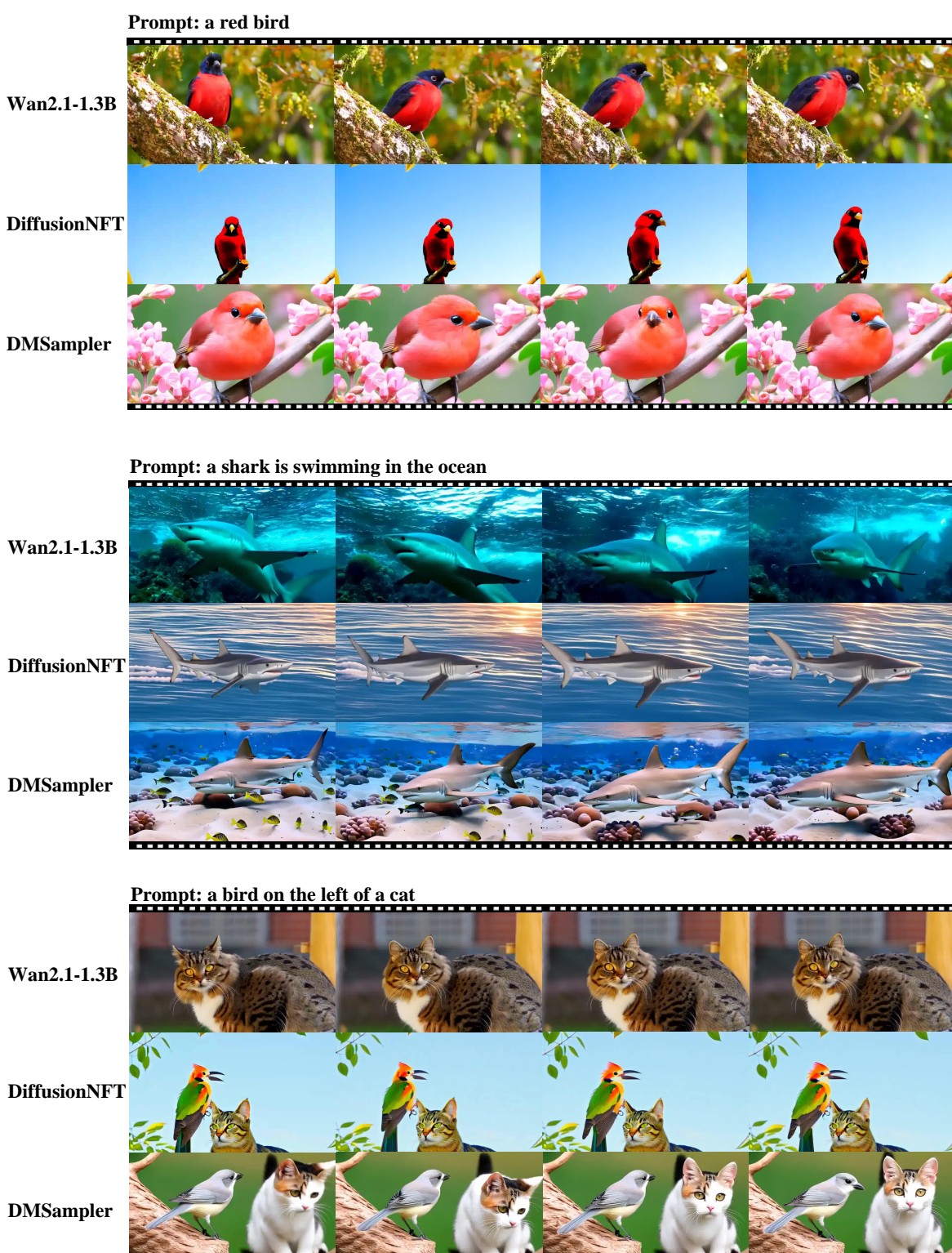

*Figure 10.* More examples of generated videos of different text-to-video models (1).

**Prompt: A boat sailing leisurely along the Seine River with the Eiffel Tower in background, black and white**

**Prompt: a bear climbing a tree**

**Prompt: A fat rabbit wearing a purple robe walking through a fantasy landscape**

*Figure 11.* More examples of generated videos of different text-to-video models (2).

**Prompt: a polar bear is playing guitar**

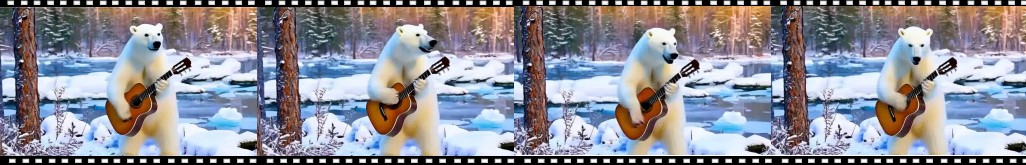

**Prompt: a dog enjoying a peaceful walk**

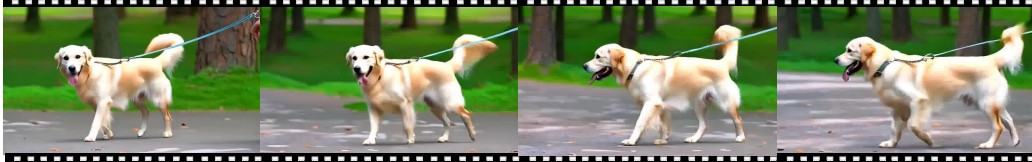

**Prompt: a person is archery**

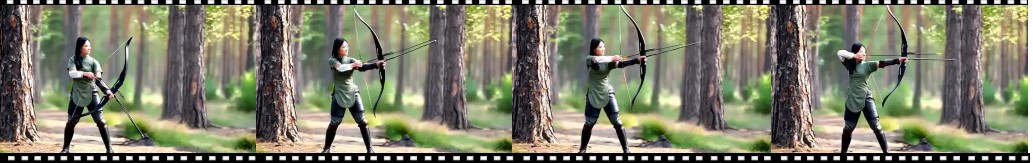

**Prompt: a person playing guitar**

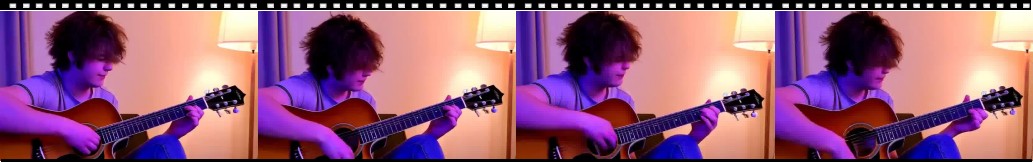

*Figure 12.* More visual examples of base model of DMSampler.

**Prompt: A panda cooking in the kitchen**

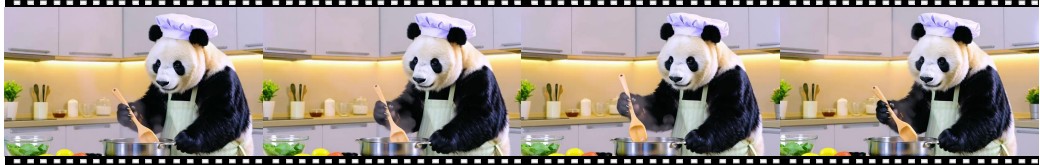

**Prompt: a cat and a dog**

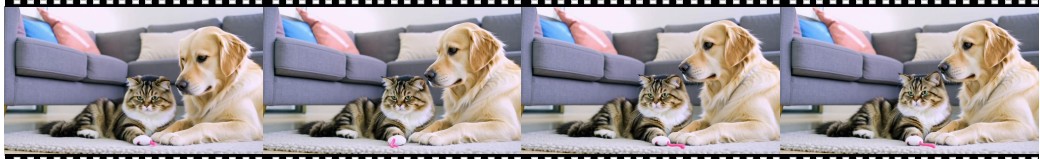

**Prompt: a person drinking coffee in a cafe**

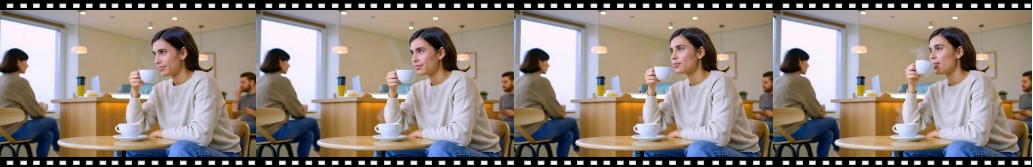

**Prompt: A person is tai chi**

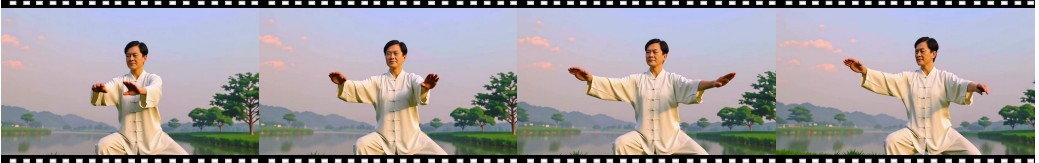

*Figure 13.* More visual examples of distillation model of DMSampler.

