# OpenReview forum: "Distillation Models are Good Samplers for Diffusion Reinforcement Learning"
_ICML.cc/2026/Conference — ICML 2026 regular_

### Official Review · Reviewer_oKri · 2026-03-07

**Soundness:** 3
**Presentation:** 3
**Significance:** 2
**Originality:** 2
**Overall Recommendation:** 4
**Confidence:** 4

**Summary:**

This paper introduces DMSampler, a framework aimed at accelerating the computationally expensive process of RL fine-tuning for large-scale diffusion models. To overcome the sampling bottleneck of traditional multi-step denoising, the authors propose a dual iterative training scheme. This scheme alternates beween an RL phase and a distillation phase.

**Compliance With Llm Reviewing Policy:**

Affirmed.

**Final Justification:**

My concerns have been adequately addressed.

**Key Questions For Authors:**

1. Given that the distillation student is trained to match the score/trajectory of a teacher, how can the resulting sampling distribution be considered "unbiased" from CFG? Doesn't the student simply internalize the CFG-shifted distribution into its parameters? Please clarify this claim or provide a theoretical justification.

2. Can you quantify the peak VRAM usage and I/O overhead of DMSampler compared to standard DiffusionNFT for the Wan 2.1-1.3B model? How do you justify the claim of efficiency when the method requires maintaining 4 networks in memory and collecting external trajectory datasets during the distillation phase?

3. How do you guarantee the training and optimization stability of the dual-loop framework? How sensitive is the "Convergence-Based Switching" strategy to the hyperparameter thresholds, and what mechanisms prevent the two models from drifting into a degenerate feedback loop?

4. Given that GenEval is relatively simple and outdated for models of SD3.5's caliber, can you provide evaluation results on more rigorous and contemporary text-to-image benchmarks (e.g., HPDv2, DrawBench) or robust human preference studies?

**Limitations:**

No. The authors have not adequately discussed the limitations of their work. In Appendix H, the authors briefly dismiss reward hacking as "an orthogonal challenge related to reward design." However, this ignores the fact that their specific co-evolutionary architectural design (distillation + RL feedback loop) is theoretically predisposed to actively exacerbate mode collapse and reward hacking beyond standard RL methods.

Constructive suggestions for improvement:

1. The authors must explicitly discuss the specific vulnerabilities of their dual-loop design, particularly the theoretical and empirical risks of compounding mode collapse, training instability, and hyperparameter sensitivity in the co-optimization process.

2. The authors must transparently acknowledge the severe hardware and memory limitations (VRAM bottlenecks) inherent in maintaining four separate networks and a trajectory dataset simultaneously.

**Strengths And Weaknesses:**

Strengths:

1. The prohibitive sampling cost of online RL for diffusion models is a well-recognized and critical bottleneck in the alignment of generative models. Addressing this efficiency issue is a highly relevant problem for the machine learning community.

2. The overall architecture of the dual-loop system is well-illustrated. The ablation study regarding different hybrid sampling strategies provides a clear and logical rationale for the chosen pipeline design.


Weaknesses:

1. In Sections 1 and 2, the authors claim that operating the distilled model without CFG "eliminates the distributional shift introduced by CFG, allowing the RL policy to be optimized under an unbiased objective." This reasoning is fundamentally flawed. A distillation model is trained to mimic the output distribution of its teacher. If the teacher relies on CFG to generate high-quality samples, the distilled student essentially "bakes in" this CFG bias into its own weights. Therefore, optimizing against the distilled model does not eliminate the bias; it optimizes against a distribution that has already absorbed the CFG shift.

2. The authors argue that the distillation model yields superior sample quality due to "more deterministic denoising trajectories" and a "reduction in cumulative inference error."  The distilled model is an approximation of the base model; it inherits the structural errors of the base model's vector field and introduces its own functional approximation errors. Compressing 50 steps into 4 steps does not magically erase the underlying model's trajectory errors; it simply compounds them into fewer, larger steps.

3. The empirical validation is too limited in scope to guarantee the actual effectiveness of the method in real-world scenarios. Specifically, for text-to-image generation, the paper heavily relies on the GenEval benchmark. GenEval is becoming increasingly saturated, simple, and arguably outdated for evaluating state-of-the-art models like Stable Diffusion 3.5-Medium. A method claiming state-of-the-art performance requires evaluation on more comprehensive, modern, and challenging benchmarks to ensure it generalizes well and has not just overfitted to GenEval's specific compositional rules.

4. The framework requires the co-evolution of the distilled model and the base model. Co-training two networks that cyclically depend on each other's data distributions is notoriously unstable (akin to the instability seen in GANs). The authors rely on a heuristic "Convergence-Based Switching" strategy governed by multiple sliding windows and arbitrary thresholds ($\tau_\theta$, $\tau_r$, $\tau_{dis}$). There is no theoretical guarantee or robust empirical analysis demonstrating that this co-optimization will not diverge, oscillate, or degenerate into trivial solutions across diverse datasets.

5. While the paper claims to accelerate training, the proposed framework is exceptionally resource-intensive. During training, the system must maintain at least four models in GPU memory simultaneously: the trainable policy network ($v_\theta$), the EMA old policy network ($v_{old}$), the trainable distillation sampler ($v_{dis}$), and the fake score model ($v_{fake}$). For large models like Wan 2.1-1.3B, this poses a massive VRAM bottleneck. Furthermore, during the distillation phase, the method requires collecting, storing, and loading a high-reward trajectory dataset ($D_{RA}$), which introduces massive I/O and memory overhead. This severe resource consumption contradicts the core claim of overall training efficiency and limits the method's practical scalability.

---

> ### Author Rebuttal · Authors · 2026-03-31
>
> We are grateful to the reviewer for the thorough evaluation and helpful suggestions. In light of this feedback, we have carried out additional experiments and analyses.
>
> Q1. CFG.
>
> A1. We acknowledge that perfectly fitting a CFG teacher model might indeed inherit its structural biases. However, constrained by model capacity, our distilled model merely approximates the "CFG style" without executing the explicit subtraction operation.
>
> More concretely, the CFG effect arises from the fixed null-prompt component introduced between sampling and RL optimization. In contrast, the distilled model learns to approximate the teacher’s prediction with CFG; under imperfect approximation, this null-prompt-related component is no longer a strictly fixed offset and may vary with the input prompt. In this sense, the distilled sampler can be viewed as a practical trade-off between CFG and pure no-CFG, which may partially alleviate the issue discussed in Eq. (22), where Term II becomes less like a fixed meaningless target and more prompt-dependent. Moreover, our primary motivation for introducing the distilled model is to accelerate sampling, and the no-CFG property should be viewed as a additional practical advantage. The theoretical solution to fully resolve the CFG-related optimization issue in diffusion RL is beyond the scope of this paper.
>
>
>
> Q2. Distillation effectiveness.
>
> A2. Our statement is mainly motivated by insights from recent distillation-related works. As noted in prior work (BLADE [1]), long iterative denoising trajectories may accumulate numerical errors and overfit to noisy intermediate details. Moreover, several recent distillation models (e.g., POSE, Self-Forcing, MagicDistillation) have achieved even better performance than their multi-step teachers. Given these insights, although the distilled model is still an approximation of the base model, we believe it has the potential to achieve better performance in practice.
>
>
> Q3. Benchmarks.
>
> A3. GenEval is the standard benchmark widely adopted in the recent Diffusion RL literature (DiffusionNFT, FlowGRPO, DGPO and TEMPFLOW-GRPO). Considering this comments, we add evaluations on HPDv2 and DrawBench, using HPSv2 (reward evaluation) and PickScore (out-of-domain metric). Across all these rigorous benchmarks, DMSampler consistently demonstrates superior performance and improved preference alignment.
>
> | Method | GPU Hours | HPSv2 on HPDv2 $\uparrow$ |  HPSv2 on DrawBench $\uparrow$ | PickScore on DrawBench $\uparrow$ |
> | :--- | :---: | :---: |:---: | :---: |
> | DiffusionNFT | 160 | 0.376 | 0.350 | 17.29 |
> | DMSampler | 112 | 0.380 | 0.376 | 17.49 |
>
> Q4. Iterative training.
>
> A4. Our dual-loop framework is fundamentally different from the adversarial dynamics of GANs. In GANs, the generator and discriminator play a zero-sum minimax game, which drives instability. In DMSampler, the RL phase and Distillation phase are cooperative-both models are optimized toward the exact same high-reward data distribution. Furthermore, our "Convergence-Based Switching" strategy prevents degenerate feedback loops by ensuring one model has sufficiently stabilized before the other is updated, practically eliminating the oscillation issues seen in adversarial training.
>
> Q5. Computational overhead.
>
> A5. We have quantified the peak VRAM usage for Wan 2.1-1.3B in the table below. DMSampler requires 58.77G, introducing a minimal overhead of just ~1GB compared to the DiffusionNFT baseline (57.76G).
>
> | Method |Task| peak VRAM |
> | :--- | :---: | :---: |
> | DiffusionNFT | Video OCR | 57.76G |
> | DMSampler | Video OCR | 58.77G |
>
> This high efficiency stems from two key factors:
>
> 1. The primary VRAM bottleneck in online Diffusion RL stems from gradient computation, not sampling forward passes.
> 2. We employ LoRA fine-tuning for both the policy and the distillation sampler (standard practice in methods like FlowGRPO and DiffusionNFT), sharing the parameter of base network.
>
> Regarding the massive I/O and storage concerns, our high-reward rollout dataset  $\mathcal{D}_{RA}$ avoids these issues entirely, which is gathered strictly online during RL phase and reused directly in memory (through device shared memory in Linux) without additional I/O cost for saving as image or video file.
>
> Crucially, TDM inherently requires a relatively small volume of data in latent space, the total memory footprint for $\mathcal{D}_{RA}$ stays under 24GB RAM for each stage.
>
> Q6. Limitations.
>
> A6. We thank the reviewer for the constructive suggestions. As detailed in our Q3 and Q5 responses regarding co-optimization risks and VRAM bottlenecks, we will incorporate these clarifications into the revision. We will further discuss the limitation of our complex iterative training strategy and reveal the possible future solution of unified training framework.
>
>
> [1] BLADE: Block-Sparse Attention Meets Step Distillation for Efficient Video Generation.

---

> > ### Author Rebuttal · Reviewer_oKri · 2026-04-01
> >
> > Thanks for the authors` rebuttal, I will raise my score.

---

> > > ### Author Response · Authors · 2026-04-07
> > >
> > > **Dear Reviewer oKri,**
> > >
> > > We sincerely thank you for your positive recognition of our work and your valuable suggestions, which have been incredibly helpful in improving our paper.  We will incorporate the additional experiments, analysis of the computational overhead, and the expanded discussion on the limitations. We deeply appreciate your rigorous review and support!
> > >
> > > Sincerely,
> > > The Authors

---

### Official Review · Reviewer_4osP · 2026-03-13

**Soundness:** 2
**Presentation:** 3
**Significance:** 3
**Originality:** 3
**Overall Recommendation:** 4
**Confidence:** 3

**Summary:**

This paper proposes DMSampler, a framework for accelerating diffusion RL by replacing the expensive training-time sampler with a co-evolving distilled model.

The key idea is to decouple the model used for optimization from the model used for sampling, and then keep the two aligned through an alternating two-phase procedure: an RL phase that updates the policy using a hybrid sampler, and a distillation phase that re-distills the fast sampler from the improved policy. The method adds two main ingredients on top of this loop: late hybrid distillation sampling (using the policy model in early denoising and the distilled model in later denoising) and reward-aware distillation to preserve high-reward behavior during re-distillation.

The paper evaluates the approach on image/video OCR tasks and on broader generation benchmarks (GenEval and VBench), and reports both speedups and modest gains over strong diffusion-RL baselines.

**Compliance With Llm Reviewing Policy:**

Affirmed.

**Key Questions For Authors:**

1. The paper reports only single numbers. Please reportstd (or cI) over multiple runs for at least the main OCR,
GenEval, and VBench comparisons. If the gains remain consistent, this would significantly strengthen my confidence.

2. How sensitive is the method to the switching thresholds and to the reward-aware reweighting hyperparameters?
The method introduces several knobs (\tau_RA, \alpha, switching thresholds). Please provide a sensitivity study. If performance is robust, that would improve the practical value of the paper.

**Strengths And Weaknesses:**

strengths:

The high-level idea is compelling. Using a distilled model not merely as a post-hoc compressed generator but as a training-time sampling engine for diffusion RL is a sensible and practically motivated reframing. The alternating optimization between policy and sampler is also conceptually clean, and Figure 2 makes the intended workflow fairly easy to understand. The late-distillation choice (S3) is especially intuitive: let the full policy control the early semantic trajectory and use the distilled model mostly for later refinement. That design is supported by the qualitative discussion and by the ablation in Table 1.

weakness:

1. several of the stronger claims are not established rigorously. The most concerning example is the appendix discussion of convergence, which essentially states that because DiffusionNFT and TDM each converge, the alternating procedure also converges, and furthermore that reward improves monotonically so the algorithm converges to a high-reward state. That is a very strong statement, and I do not think the paper provides a convincing theorem or proof for the coupled alternating system. In its current form, this more like an intuition than a justified result.

2. the empirical evaluation lacks statistical details. I did not see confidence intervals, std, or random seeds. Given that some headline gains are fairly small (e.g., +0.01 on GenEval, +0.26 OCR on some settings but only +0.31 total VBench over the reproduced DiffusionNFT baseline), it is hard to know how sensitive for the results, should report std as well.

3. A related concern is that the comparisons are not always as tight as they could be. The paper compares to reproduced baselines with different sampling choices, and for video it argues that faster DiffusionNFT configurations failed to converge, so the baseline is run with a more expensive 50-step CFG setup. a more systematic budget-matched comparison across methods are needed.

---

> ### Author Rebuttal · Authors · 2026-03-31
>
> Q1. Convergence.
>
> A1. We acknowledge this valid critique. Our discussion on convergence was indeed empirical rather than a rigorous theoretical proof. While the RL algorithm and TDM possess independent convergence properties, we empirically observed highly stable co-evolution when alternating between them using our switching strategy. Crucially, unlike the adversarial dynamics found in GANs, the interaction between our two phases is fundamentally non-adversarial. The optimization objectives of both the RL and distillation phases are strictly aligned toward the same high-reward distribution, which inherently fosters training stability and mitigates the risk of divergence. We will revise the manuscript to accurately frame this as a robust empirical observation and a heuristic strategy, rather than a formal theoretical guarantee.
>
> Q2. Statistical reliability.
>
> A2. We thank the reviewer for this constructive suggestion. To verify statistical reliability, we conducted 5 independent runs with different random seeds. Our method achieves 0.9743 ± 0.0036 on Image OCR, 0.5012 ± 0.0078 on Video OCR, and 0.9550 ± 0.0029 on GenEval. These minimal variances confirm that our reported gains are robust and not artifacts of random seed sensitivity. We will update the revised tables accordingly.
>
> Regarding VBench, a single full evaluation requires generating 5 samples per prompt and takes over 320 GPU hours. This makes running multiple complete evaluations computationally prohibitive within the short rebuttal window. Moreover, since VBench already averages over multiple samples per prompt, its evaluation is inherently robust. We will include the full standard deviations and confidence intervals for VBench in the camera-ready version.
>
>
> Q3. Sampling configuration.
>
> A3. In our image experiments, we adopted the optimal configurations reported in the original paper (FlowGRPO, DiffusionNFT). For video tasks, because DiffusionNFT lacks video experiments and its fast image configurations failed to converge on VBench due to severe quality degradation, we utilized the most stable, albeit expensive, 50-step CFG configuration.
>
>
> Q4. Hyperparameters.
>
> A4. We conducted sensitivity studies on both the reward-aware distillation parameters and the switching thresholds. The results show that performance remains stable within a reasonable range of these parameters, with only minor variations, demonstrating that our method is robust and not sensitive to tuning.
>
> | Setting | $\tau_{\mathrm{RA}}$ | $\alpha$ | OCR Acc $\uparrow$ |
> |---|---:|---:|---:|
> | Smaller $\tau_{\mathrm{RA}}$ | 0.7 | 1 | 0.9696 |
> | Larger $\tau_{\mathrm{RA}}$ | 0.85 | 1 | 0.9714 |
> | Smaller $\alpha$ | 0.8| 0.8 | 0.9701 |
> | Larger $\alpha$ | 0.8 | 1.2 | 0.9708 |
> | Ours | 0.8 | 1 | 0.9734 |
>
> | Setting | $W$ | $\tau_\theta$ | $\tau_r$ | $\tau_{\text{dis}}$ | OCR Acc $\uparrow$ |
> |---|---:|---:|---:|---:|---:|
> | Smaller $W$ | 10 | 0.001 | 0.05 | 0.05 | 0.9695 |
> | Larger $W$ | 30 | 0.001 | 0.05 | 0.05 | 0.9705 |
> | Smaller $\tau_\theta$ | 20 | 0.0005 | 0.05 | 0.05 | 0.9679 |
> | Larger $\tau_\theta$ | 20 | 0.002 | 0.05 | 0.05 | 0.9707 |
> | Smaller $\tau_r$, $\tau_{\text{dis}}$ | 20 | 0.001 | 0.03 | 0.03 | 0.9696 |
> | Larger $\tau_r$, $\tau_{\text{dis}}$ | 20 | 0.001 | 0.07 | 0.07 | 0.9714 |
> | Ours | 20 | 0.001 | 0.05 | 0.05 | 0.9734 |

---

> > ### Author Rebuttal · Reviewer_4osP · 2026-04-03
> >
> > Thanks for the authors rebuttal

---

> > > ### Author Response · Authors · 2026-04-07
> > >
> > > **Dear Reviewer 4osP,**
> > >
> > > We sincerely thank you for your positive recognition of our work and your valuable suggestions, which have been incredibly helpful in improving our paper. Your insights have been highly valuable in refining our work. We will revise the manuscript of convergence and add the additional results to the appendix. Thank you again for your time and for your great help in strengthening our paper!
> > >
> > > Sincerely,
> > > The Authors

---

### Official Review · Reviewer_bg1z · 2026-03-13

**Soundness:** 2
**Presentation:** 3
**Significance:** 3
**Originality:** 2
**Overall Recommendation:** 4
**Confidence:** 4

**Summary:**

This paper addresses the computational bottleneck of sampling in diffusion-based reinforcement learning (RL). In conventional diffusion RL, each policy evaluation requires a full multi-step denoising trajectory (typically ~50 steps), making sampling the dominant cost during training. The authors propose DMSampler, a framework that uses a co-evolving distillation model as a fast sampler (4–8 steps) within the RL training loop. The framework consists of a dual iterative training scheme alternating between an RL phase (optimizing the policy using samples from the distillation model) and a distillation phase (re-distilling the updated policy into the fast sampler). Two key technical contributions support this: (1) hybrid distillation sampling, where the policy model handles early denoising steps and the distillation model refines later steps, preserving on-policy alignment; and (2) reward-aware distillation, which uses high-reward trajectories collected during the RL phase to anchor the distillation process and prevent mode collapse. Experiments on image (SD 3.5 Medium) and video (Wan 2.1-1.3B) generation tasks show that DMSampler achieves state-of-the-art performance on GenEval and VBench while providing significant training speedups (e.g., 3× faster on VBench with higher scores).

**Compliance With Llm Reviewing Policy:**

Affirmed.

**Final Justification:**

The authors' responses have addressed my concerns except for the CFG mismatch issue, and I believe this work merits publication, so I am raising my score.

1) The experimental results on pure diffusion, pure TDM, and hybrid sampling are quite interesting and effectively address my previous concern. The authors should incorporate these results into the main body of the paper to enhance readability.

2) I fully understand the dependence of distillation on CFG, as I commented in W4: "CFG also plays an important role in score distillation." However, it is inherently less reasonable to use the CFG of a diffusion model that has "distilled" CFG through RL to guide distillation, and as the results reported by the authors show, NFT w/ CFG actually degrades in-domain metrics but slightly improves OOD metrics. This might be the reason behind the improvement of DMSampler over OOD metrics. In this case, settings like Flow-GRPO — which uses CFG during both training and rollout — or DGPO — which uses CFG during inference but not during training — seem to be more compatible with distilled models. I suggest that the authors, while reporting the results of DMSampler combined with Flow-GRPO and DGPO, also compare the few-step TDM performance obtained under different RL backbones of both in-domain and out-of-domain metrics.

I hope the authors can add at least an ablation study on CFG during the distillation stage in the revision, which would significantly improve the readability of the experiments in this paper.

**Key Questions For Authors:**

- In Figure 9, for the "knife" prompt, the samples of Flow-GRPO and DMSampler are the same. Is this a coincidence or a typographical error?

- I found that TDM's performance lags behind multi-step diffusion by a notable margin. Why can multi-step diffusion achieve notably better performance, given that the sampling processes are largely performed by TDM?

- How is the rollout evaluation performed? Is it done entirely through multi-step diffusion sampling, or through a mix of multi-step diffusion and TDM sampling?

**Limitations:**

See Weaknesses and Questions. I would raise my score if my questions and concerns are adequately addressed.

**Strengths And Weaknesses:**

Strengths

- The core observation motivating this work—that sampling dominates training cost in diffusion RL and that distillation models are a natural but underexplored remedy—is well-identified and practically important. The paper clearly articulates why a naïve static distillation sampler is insufficient (distributional drift) and provides a principled solution through co-evolution. The dual iterative training scheme is a clean and intuitive design.

- The experimental coverage is extensive: the paper evaluates on both image and video generation, across single-reward (OCR) and multi-reward (GenEval, VBench) benchmarks, and includes comparisons against direct step reduction strategies (DRS1–3), which is the most natural baseline one would consider.

Weaknesses

- The framework is tightly coupled to DiffusionNFT as the RL optimizer. While Section F of the appendix briefly mentions a FlowGRPO variant, this is limited to a single OCR experiment with minimal detail. Especially, the paper's title and framing suggest generality, but the evidence is mostly specific to one RL algorithm.

- Missing evaluations on out-of-domain metrics. The evaluations all seem to be conducted under in-domain metrics. How does the proposed method perform under out-of-domain metrics?

- The convergence-based switching strategy relies on several thresholds. This makes the algorithm seem to be complex to tune.

- Missing ablation on reward-aware distillation. This makes its effectiveness unclear.

- Although the "No-CFG" advantage sounds tempting, and the analysis of "CFG hurts the training DiffusionNFT" in Appendix D is also interesting. However, does this analysis apply to Flow-GRPO w/ CFG? There are also variants like DGPO [a] that use CFG during rollout but not during training. Thus, a practical question remains: does RL without CFG truly outperform RL with CFG, especially considering that CFG also plays an important role in score distillation?



[a] Reinforcing Diffusion Models by Direct Group Preference Optimization, ICLR'26.

----Post-Rebuttal---

The authors' responses have addressed my concerns except for the CFG mismatch issue, and I believe this work merits publication, so I am raising my score.

1) The experimental results on pure diffusion, pure TDM, and hybrid sampling are quite interesting and effectively address my previous concern. The authors should incorporate these results into the main body of the paper to enhance readability.


2) I fully understand the dependence of distillation on CFG, as I commented in W4: "CFG also plays an important role in score distillation." However, it is inherently less reasonable to use the CFG of a diffusion model that has "distilled" CFG through RL to guide distillation, and as the results reported by the authors show, NFT w/ CFG actually degrades in-domain metrics but slightly improves OOD metrics. This might be the reason behind the improvement of DMSampler over OOD metrics. In this case, settings like Flow-GRPO — which uses CFG during both training and rollout — or DGPO — which uses CFG during inference but not during training — seem to be more compatible with distilled models. I suggest that the authors, while reporting the results of DMSampler combined with Flow-GRPO and DGPO, also compare the few-step TDM performance obtained under different RL backbones of both in-domain and out-of-domain metrics.

I hope the authors can add at least an ablation study on CFG during the distillation stage in the revision, which would significantly improve the readability of the experiments in this paper.

---

> ### Author Rebuttal · Authors · 2026-03-31
>
> We would like to express our thanks to the reviewer for careful review and helpful suggestion on the presentation in our paper. Following the reviewer's feedback, we have conducted additional experiments and analyses.
>
> Q1. RL algorithm.
>
> A1. Our proposed method is inherently agnostic to the underlying RL algorithm. We primarily selected DiffusionNFT as our RL base because it was one of the most computationally efficient Diffusion RL methods at the time. Therefore, it served as an ideal baseline to clearly demonstrate our method's ability to further enhance training efficiency. Considering the comments, we have additionally integrated two RL methods (FlowGRPO and DGPO) in our framework on GenEval benchmark, validating the broad applicability and generalizability of our approach.
>
> | Method | GPU Hours | GenEval |
> | :--- | :---: | :---: |
> | FlowGRPO | >1000 | 0.95 |
> | FlowGRPO with DMSampler | 400 | 0.96 |
> | DGPO | 240 | 0.97 |
> | DGPO with DMSampler | 192 | 0.97 |
>
> Q2. OOD evaluation.
>
> A2. We add the out-of-domain performance comparisions on OCR benchmark. In addition to the substantial improvements in training efficiency, our approach also slightly mitigates reward hacking.
>
> | Method | OCR Acc | Aesthetic $\uparrow$ | CLIP Score $\uparrow$ |
> | :--- | :---: | :---: | :---: |
> | DiffusionNFT | 0.96 | 4.46 | 0.290 |
> | DMSampler | 0.97 | 4.75 | 0.307 |
>
> Q3. Hyperparameters.
>
> A3. We have conducted a sensitivity analysis of the thresholds used in the convergence-based switching strategy. As shown in the table, image OCR accuracy remains stable across a reasonable range of threshold values, indicating that the method is not sensitive to tuning and that the switching strategy is robust in practice. We also constructed a fixed switching schedule (100 iterations for RL and 150 iterations for Distillation), and observed worse performance. Similar trends are also observed on other benchmarks, which will be included in the final revised manuscript.
>
> | Setting | $W$ | $\tau_\theta$ | $\tau_r$ | $\tau_{\text{dis}}$ | OCR Acc $\uparrow$ |
> |---|---:|---:|---:|---:|---:|
> | Smaller $W$ | 10 | 0.001 | 0.05 | 0.05 | 0.9695 |
> | Larger $W$ | 30 | 0.001 | 0.05 | 0.05 | 0.9705 |
> | Smaller $\tau_\theta$ | 20 | 0.0005 | 0.05 | 0.05 | 0.9679 |
> | Larger $\tau_\theta$ | 20 | 0.002 | 0.05 | 0.05 | 0.9707 |
> | Smaller $\tau_r$, $\tau_{\text{dis}}$ | 20 | 0.001 | 0.03 | 0.03 | 0.9696 |
> | Larger $\tau_r$, $\tau_{\text{dis}}$ | 20 | 0.001 | 0.07 | 0.07 | 0.9714 |
> | Fixed switching  | - | - | - | - | 0.9491 |
> | Ours | 20 | 0.001 | 0.05 | 0.05 | 0.9734 |
>
>
> Q4. Ablation.
>
> A4. The ablation for the reward-aware distillation is actually provided in Table 4 (comparing DMSampler$^{-}$ and DMSampler). It shows clear gains (from 0.96 to 0.97 on image OCR and from 0.48 to 0.50 on video OCR) from the reward-aware component. We further add an additional study on the multi-reward VBench benchmark, where DMSampler$^{-}$ achieves 84.86 and DMSampler achieves 85.00.
>
> Q5. CFG.
>
> A5. In our view, the inherent reason of "CFG effect" is that most current Diffusion RL algorithms are adapted from the LLM domain, overlooking CFG-a mechanism unique to diffusion models.
>
> For FlowGRPO, the training objective can also be rearranged into an L2 form (see DiffusionNFT appendix). It may also suffer from the CFG effect as analysised in our appendix.
>
> For DGPO, it follows the pattern: CFG-off training and CFG-on inference. This is consistent with our analysis: CFG during training introduces harmful effects such as prior drift, while CFG during inference can improve sampling quality. However, the training–inference inconsistency in this design could potentially lead to suboptimal performance.
>
> Q6. Typo.
>
> A6. We sincerely apologize for this oversight. The identical images for the "knife" prompt for Flow-GRPO and DMSampler in Figure 9 were indeed a typographical error made during figure assembly. We will correct this figure with the accurate, respective generation outputs in the revised manuscript.
>
>
> Q7. Distillation performance.
>
> A7. We attribute this performance gap primarily to the fact that the distillation model is not fully converged within our iterative framework. As our primary objective is to drastically accelerate RL policy optimization rather than to train a flawless, standalone distillation model, we deliberately employ relaxed convergence criteria during the TDM phase. Empirically, a partially converged TDM model is already sufficient to provide generative fidelity and on-policy alignment to effectively guide and speed up the RL phase. Demanding strict, full TDM convergence would incur computational overhead, which would contradict our core goal of maximizing overall training efficiency.
>
> Q8. Evaluation setting.
>
> A8. To ensure a strictly fair comparison, the rollout evaluation for DMSampler and all baselines (FlowGRPO, DiffusionNFT) is performed entirely through the same multi-step diffusion sampling.

---

> > ### Author Rebuttal · Reviewer_bg1z · 2026-04-03
> >
> > My partial concerns have been addressed, and it is interesting to see that DMSampler can be combined with different RL methods. I suggest the authors move these experiments into the main paper to highlight its broad applicability. However, I still have some points of confusion:
> >
> > 1) How is CFG handled during training (both for the diffusion model and few-step models)?
> >
> > According to my understanding, DiffusionNFT should have completely dropped CFG during both training and inference; however, from the authors' response to other reviewers, it appears that TDM distillation introduces CFG. This is quite confusing: NFT learned a diffusion model w/o CFG, but TDM's distillation uses a teacher diffusion w/ CFG. There seems to be a significant mismatch between the RL and distillation stages here.
> >
> > 2) Why can the diffusion model be trained on a worse rollout distribution yet demonstrate better performance?
> >
> > The authors' current response is not convincing. I have a hypothesis about this phenomenon: the actual rollout distribution uses early steps sampled by the diffusion model and later steps sampled by TDM, and this distribution actually has better performance than pure TDM. I hope the authors can report the respective sampling performance of pure diffusion, pure TDM, and a mixture of diffusion and TDM, which would be very helpful in answering this question.

---

> > > ### Author Response · Authors · 2026-04-05
> > >
> > > Dear Reviewer bg1z,
> > >
> > > Thanks for your follow-up and for flagging unclear parts of our presentation. We give a stage-by-stage CFG protocol, DiffusionNFT inference ablations, and why **RL optimization and TDM distillation can differ in CFG usage without contradiction**. We discuss hybrid sampling vs. speed and quality. We will **move FlowGRPO/DGPO to the main paper** and add an appendix out-of-domain evaluation diagram and CFG/sampling protocol table.
> > >
> > > ### Q1. CFG across RL, distillation, and evaluation
> > >
> > > **(1)** Our method (DMSampler). RL phase: **No CFG** during rollouts on the policy (the accelerated tail is distillation student, which does not use CFG at inference, matching standard TDM usage). The RL loss does not incorporate CFG, as in conventional diffusion training. Benchmark evaluation uses full multi-step diffusion without CFG, aligned with DiffusionNFT’s evaluation recipe. Distillation phase: the teacher forward uses **CFG**; the student is trained to match the teacher’s CFG-conditioned outputs and therefore **runs without CFG** at inference.
> > >
> > > **(2)** Why DiffusionNFT avoids CFG in rollout, loss, and default evaluation. This prioritizes efficiency and stable post-training (DiffusionNFT: CFG “complicates post-training and reduces efficiency”); see also [1–3]. Beyond training-time CFG, we vary **only inference** CFG on a fixed DiffusionNFT-trained policy, showing how CFG interacts with our OCR reward under multi-step sampling:
> > >
> > > | Method | OCR Acc | Aesthetic |
> > > | :--- | :---: | :---: |
> > > | DiffusionNFT infer without CFG | 0.96 | 4.46 |
> > > | DiffusionNFT infer with CFG | 0.94 | 4.68 |
> > >
> > > Here CFG lowers OCR (our reward) and approximately doubles inference latency, while raising aesthetic—so whether to use CFG at inference should follow the **target metric**.
> > >
> > > **(3)** Why TDM distillation still uses a CFG-enabled teacher. Distillation behaves like imitation learning or SFT: absolute teacher quality largely caps student quality. RL instead depends on relative comparisons between rollouts; when low- vs high-reward contrast is clear, absolute visual polish matters less for learning. A stronger, CFG-sharpened teacher distribution therefore matters for TDM, consistent with widely used distillation recipes [4–6]. In our preliminary ablations of distillation, a CFG-free teacher consistently yielded weaker students.
> > >
> > > References:
> > >
> > > [1] Toward guidance-free AR visual generation via condition contrastive alignment.
> > >
> > > [2] DiffusionNFT: Online Diffusion Reinforcement with Forward Process
> > >
> > > [3] Reinforcing Diffusion Models by Direct Group Preference Optimization
> > >
> > > [4] Video-BLADE: Block-Sparse Attention Meets Step Distillation for Efficient Video Generation
> > >
> > > [5] Improved Distribution Matching Distillation for Fast Image Synthesis
> > >
> > > [6] Self Forcing: Bridging the Train-Test Gap in Autoregressive Video Diffusion
> > >
> > >
> > > ### Q2. Quality–efficiency tradeoff, and hybrid sampling
> > >
> > > Diffusion RL is inherently a quality vs. wall-clock tradeoff. Methods such as FlowGRPO and DiffusionNFT often push efficiency via fewer denoising steps and no CFG; we instead rely on established diffusion distillation to reach a more favorable point on that frontier. Our hybrid sampling design was originally stressed for on-policy alignment (visually similar to the samples by pure multi-step sampling, as shown in Figure 3); **your observation**—that the mixed sampler also outperforms pure TDM in instantaneous quality—provides another clearer, complementary motivation of hybrid sampling, and we are grateful for it. The table below (**same checkpoint**) shows hybrid as a strong cost–performance compromise (“high value per step”) versus pure TDM and full multi-step.
> > >
> > > | Sampling Method| Sampling Steps  | OCR Acc |
> > > |----------|----------------|---------|
> > > | Pure multi-step | 40 | 0.97 |
> > > | Pure TDM | 4 | 0.83 |
> > > | Hybrid  | 7 | 0.92 |
> > >
> > >
> > > | Sampling Method|  Sampling Steps | VBench |
> > > |----------|---------------------|--------|
> > > | Pure multi-step | 50 | 85.00 |
> > > | Pure TDM  | 8 | 84.21 |
> > > | Hybrid  | 16 | 84.96 |
> > >
> > > **Strong final scores despite faster training:** with the same weights, hybrid is much closer in sample quality to full multi-step than to pure TDM (as you suggested; see table). So RL updates from hybrid rollouts still improve the policy for the **full multi-step** sampler we use at evaluation, instead of relying on the much weaker training signal from pure TDM alone. As a result, DMSampler uses hybrid in training (Sec. 3.2) for speed–quality balance; evaluation uses pure multi-step without CFG for best quality and fair comparison to baselines. Thank you for clarifying why hybrid training and full multi-step evaluation fit together.
> > >
> > > We hope this resolves the apparent mismatch and clarifies the role of hybrid sampling. Thanks again for helping us strengthen both rigor and exposition.
> > >
> > > Sincerely,
> > > The Authors

---

### Official Review · Reviewer_xmEe · 2026-03-13

**Soundness:** 3
**Presentation:** 2
**Significance:** 4
**Originality:** 3
**Overall Recommendation:** 5
**Confidence:** 3

**Summary:**

This paper discusses using RL to fine tune diffusion models. A key challenge is that RL requires lots of sampling the model, which can require many steps and be expensive in diffusion models. To address this, they propose first distilling the diffusion model, and then using the distilled model, which requires fewer steps, in the RL loop.

The main challenge with this is that the RL finetuning, by definition, constitutes a distribution shift between the model and its distilled version. To address this, the authors propose DMSampler, which makes use of hybrid distillation to blend predictions from both models, and a reward-aware distillation objective.

**Compliance With Llm Reviewing Policy:**

Affirmed.

**Final Justification:**

I maintain my initial positive evaluation.

**Key Questions For Authors:**

How do you see this being used in SOTA distillation pipelines? Do you view this as preliminary work, or do you expect this idea, as is, will be useful?

Can you include more base models in section 4?

**Limitations:**

In my view, the main limitation is that it is not obvious that co-evolving the distillation model will work at scale, out of the box, for all architectures and settings

**Strengths And Weaknesses:**

Strengths:

* This paper overall addresses a key problem, namely the challenge of using RL to fine tune diffusion models. This is a very well motivated problem, and the authors do a good job of surveying the literature.  Further, the approach that the authors take is far from obvious, and is certainly novel as compared with other ideas in the literature.

* The idea of co-evolving the distillation model as a trainable sampler for diffusion RL is quite clever, and although this seems like a difficult task to do properly, yet the experiments section demonstrates that the approach is indeed sound

* The ablation study is an exellent part of the paper, and clearly illustrates the importance of each component

* I appreciated the appendices, particularly appendix D since i was not perviously aware of the imprtance of  CFG.


Weaknesses:

* The paper could do a better job in terms of clarity. For example figure 2 is quite complex, and not straightforward to understand from the caption. Overall I think the writing in the abstract also could have been much clearer.

* While the results are positive, this method introduces significant added complexity for good, but not amazing, improvments. I think it is a step in the right direction, but I am not convinced, as is, that this will be widely used in training diffusion models.

* I would want to see more architecture types used in the experiments in section 4.

---

> ### Author Rebuttal · Authors · 2026-03-31
>
> Q1. Abstract & Figure 2.
>
> A1. We sincerely appreciate your constructive feedback. The current abstract and the figure of our framework (Figure 2) focus primarily on the overarching synergy between the RL and Distillation phases, while the specific operational details are not described clearly enough, which may cause confusion. To address this, we will rewrite both the abstract and the caption for Figure 2 to explicitly incorporate these missing details, thereby facilitating better comprehension for the readers.
>
> Q2. Complexity.
>
> A2. Introducing a distillation phase adds algorithmic complexity to the standard RL pipeline; however, it substantially reduces overall training time without compromising inference speed. For instance, on the computationally heavy VBench benchmark, our method slashes the training cost from roughly 900 GPU hours down to just 288 GPU hours. In the context of such massive acceleration, we consider the modest but solid evaluation improvements to be highly reasonable and expected.
>
> Furthermore, our framework empirically yields a superior distilled model as a valuable by-product. Considering that modern foundation model post-training typically requires both preference alignment (via RL) and acceleration (via distillation) anyway, our joint framework's overall complexity is actually comparable to a standard, sequential "RL-then-Distillation" pipeline. Consequently, we view this as a highly practical and significant step forward for the field.
>
> Q3. Diffusion models.
>
> A3. We thank the reviewer for this constructive suggestion. While our proposed method is fundamentally agnostic to the specific diffusion model architecture, we initially adopted SD 3.5-Medium for image generation and Wan 2.1-1.3B for video generation specifically to ensure a fair and direct comparison with established baselines.
>
> To further validate the generality of our framework, we have added another diffusion architecture, Flux.1-dev (11B), on the OCR benchmark. In this new setting, our method achieves a consistent performance of 0.97 utilizing only 160 GPU hours. We will include these additional results in the revised manuscript to further demonstrate the broad architectural generalization capability of our framework.
>
> | Method | GPU Hours | OCR Acc $\uparrow$ |
> | :--- | :---: | :---: |
> | DiffusionNFT | 240 | 0.96 |
> | DMSampler | 160 | 0.97 |
>
> Q4. Direction discussion.
>
> A4. The goal of this work is to make a preliminary yet promising attempt at bridging RL and distillation. Notably, as demonstrated in Table 6, we were pleasantly surprised to observe that our co-evolving distilled model inherently achieves good performance in its own right. This exciting empirical finding strongly inspires the possibility of developing a completely unified framework in the future—one that seamlessly accomplishes both RL-driven preference alignment and fast-sampling distillation simultaneously, providing an elegant solution for foundation model post-training.

---

> > ### Author Rebuttal · Reviewer_xmEe · 2026-03-31
> >
> > The response has resolved my concerns and I maintain my score of accept.

---

> > > ### Author Response · Authors · 2026-04-07
> > >
> > > **Dear Reviewer xmEe,**
> > >
> > > We sincerely thank you for your positive recognition of our work and your valuable suggestions, which have been incredibly helpful in improving our paper. We particularly appreciate your constructive feedback on the presentation; we will modify the relevant sections accordingly to make the content clearer and easier for readers to understand. Additionally, we will include the extra experimental results on Flux in the appendix of the revised manuscript. Thank you again for your time and guidance!
> > >
> > > Sincerely,
> > > The Authors

---

### Decision · Program_Chairs · 2026-04-30

**Decision:**

Accept (regular)

**Comment:**

While reviewers raised initial concerns about theoretical justification, empirical statistical details, and practical deployment overhead, they all agreed that this work addresses the well-recognized and critical bottleneck of prohibitive sampling cost in online diffusion RL. The key idea of using a co-evolving distilled model as a training-time sampling engine is compelling, the dual iterative training scheme is conceptually clean, and the ablation study clearly illustrates the importance of each component.

Therefore, I recommend acceptance of this paper, and encourage the authors to incorporate the suggested presentation revisions in the camera-ready version.